# Rethinking Forgery Attacks on Semantic Watermarks in Black-Box Settings: A Geometric Distortion Perspective

**Cheng-Yi Lee** [1]  **Yichi Zhang** [2]  **Yuchen Yang** [2]  **Chun-Shien Lu** [1]  **Jun-Cheng Chen** [1]

## Abstract

Recent studies have shown that semantic watermarks, which embed information into the initial noise of latent diffusion models (LDMs), are vulnerable to black-box forgery attacks. However, existing methods primarily rely on empirical evidence and lack a rigorous theoretical understanding of the conditions under which such attacks succeed or fail. To bridge this gap, we rethink the nature of such attacks through the lens of rate-distortion in the latent space. Our analysis identifies an irreducible distortion floor due to structural mismatches between proxy and target models, which fundamentally limits the fidelity of forged watermarks. We further characterize this distortion as structured geometric deviations on the latent manifold, in the form of global drift and local deformation rather than stochastic noise. Leveraging these insights, we propose a scheme-agnostic detection method that distinguishes forged samples before watermark verification. Extensive experiments demonstrate the effectiveness of our method across diverse black-box scenarios, while preserving robustness to common distortions.

## 1. Introduction

The increasing proliferation of AI-generated content (AIGC) (Cao et al., 2025) has attracted widespread interest across various fields and contributed to substantial commercial value (Betker et al., 2023; Midjourney, Inc., 2025). In visual content generation, diffusion models (Ho et al., 2020; Song et al., 2021; Rombach et al., 2022) allow individuals from diverse backgrounds to produce high-quality images with minimal effort. However, this advancement has raised concerns regarding the erosion of trust in digital

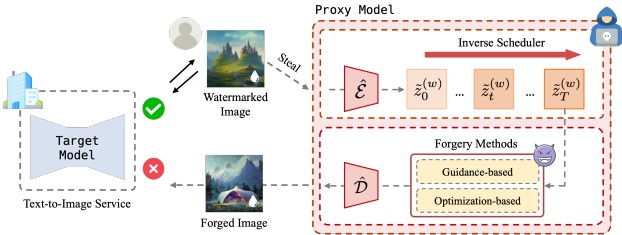

*Figure 1.* Illustration of the black-box forgery attack. An adversary uses a proxy model to invert a watermarked image into a latent representation and generates a forged image via different strategies, which preserves the service provider's watermark while falsifying content provenance.

media (Goodman, 2024) and the dissemination of misinformation (Jaidka et al., 2025). For instance, deepfakes (Westerlund, 2019), highly realistic AI-generated media, have been used to perpetrate fraud, damage personal reputations, and spread disinformation. In response, governments (Biden, 2023; Legislature, 2024; Union, 2024) have begun mandating that companies ensure the detectability and traceability of generated images or their underlying models, as well as the identification of responsible users.

Generative image watermarking has emerged as a promising paradigm to address these challenges. By injecting watermarks into generated images, the service provider (SP) can enable reliable detection and source attribution through watermark extraction. However, post-hoc watermarking schemes (Cox et al., 2008; Bui et al., 2023; Sander et al., 2025) alter the data distribution and degrade visual fidelity, regardless of whether operating in the spatial (Li et al., 2009) or frequency domain (Al-Haj, 2007). To balance utility and robustness, the *semantic watermarks* (Wen et al., 2023; Yang et al., 2024b; Gunn et al., 2025) modify the initial latent noise to embed a predefined pattern, which can later be recovered through inversion of the denoising process. This design enables straightforward deployment into existing diffusion models, achieving greater robustness against diverse image transformations and adversarial attacks.

Despite these advantages, recent studies (Müller et al., 2025) reveal that an adversary can perform a watermark forgery attack using only black-box access to the SP's model. Fig. 1 illustrates this attack. The adversary takes a watermarked

[1]Academia Sinica, Taipei, Taiwan, ROC [2]The Pennsylvania State University, PA, US. Correspondence to: Jun-Cheng Chen <pullpull@citi.sinica.edu.tw>.

*Proceedings of the 43rd International Conference on Machine Learning*, Seoul, South Korea. PMLR 306, 2026. Copyright 2026 by the author(s).

image generated by an SP, inverts it into the latent space using a proxy diffusion model, and regenerates a new image such that the SP's watermark is preserved while the image content is no longer produced by the SP. Such an attack undermines trust in the watermarking system by falsely attributing image provenance to an SP and wrongly accusing regular users of originating harmful content. However, prior work primarily demonstrates such attacks empirically, with limited insight into the conditions under which black-box forgery succeeds or fails. As a result, it remains unclear how much security semantic watermarking systems can provide in practice, or how to design defenses that generalize beyond specific attack implementations.

In this paper, we revisit the feasibility of black-box forgery attacks on semantic watermarks. We posit that the success of such attacks is constrained by distortions introduced in the latent space when the adversary relies on a mismatched proxy model. To formalize this intuition, we model black-box forgery as a rate–distortion problem, where the adversary must trade off successful watermark information transfer (rate) against the preservation of latent quality (distortion). Under this framework, we show that proxy–target model mismatch induces an irreducible distortion floor (see Fig. 2), which fundamentally limits the adversary's ability to achieve high-fidelity forgery. Crucially, we find that this distortion is not stochastic noise, but manifests as structured geometric deviations from the intrinsic latent manifold, characterized by global drift and local deformation. By leveraging these geometric differences, forged samples can be detected as a pre-verification step, without requiring any modification to existing watermarking schemes. Experimental results validate the effectiveness of our approach across a wide range of black-box forgery settings. Our contributions can be summarized as follows:

• To the best of our knowledge, we are the first to formalize black-box forgery attacks within a rate–distortion framework, identifying an irreducible distortion floor that fundamentally constrains the adversary's capability.

• We characterize forgery-induced distortion as structured geometric deviations on the intrinsic latent manifold, manifesting as global drift and local deformation.

• We propose a scheme-agnostic detection based on these geometric findings. Extensive evaluations demonstrate its effectiveness under diverse black-box forgery scenarios.

## 2. Related Works

**Semantic Watermarking.** Semantic watermarks (Wen et al., 2023; Yang et al., 2024b; Lee & Cho, 2025) embed or map a specific, recoverable structure into the starting latent, from which the diffusion process generates watermarked images. During extraction, the denoising inversion

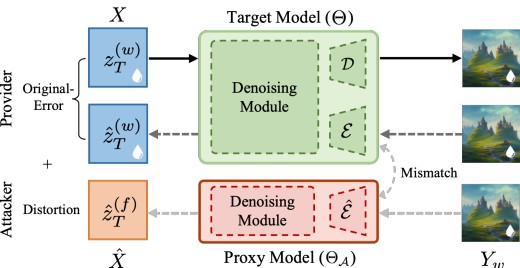

*Figure 2.* Latent distortion under model mismatch. Discrepancies between the target model ($\Theta$) and the proxy model ($\Theta_{\mathcal{A}}$) exacerbate reconstruction distortion in the latent space, arising from both intrinsic and external errors. (See Sec 4.2 for more details.)

process is applied to the watermarked image to recover the corresponding watermarked latent. This approach is highly effective because it is easy to implement without additional training. For example, Tree-Ring (TR) (Wen et al., 2023) embeds circular patterns into the frequency domain of the latent $z_T^{(w)}$. For detection, it verifies the pattern by checking if the frequency representation of $\hat{z}_T^{(w)}$ is sufficiently close to the original pattern. Gaussian Shading (GS) (Yang et al., 2024b) combines stream cipher encryption with distribution-preserving sampling to ensure that watermarked images follow the same distribution as non-watermarked ones. During verification, this process is inverted to recover a bit string, which is compared against registered keys. However, recent works (Müller et al., 2025) show that an adversary can readily forge a watermarked pattern using any arbitrary proxy model through reprompting and imprint attacks. Further details on semantic watermarks are provided in Sec. A.1.

**Rate-Distortion Theory.** Shannon first explored the essential balance between the minimum amount of information (rate) required to represent a source and the distortion that arises when the data is reconstructed (Shannon, 1948; Shannon et al., 1959). By establishing theoretical limits on compression performance, rate–distortion (RD) theory guides the design of practical source coding schemes (Ballé et al., 2020) and enables evaluation of their capabilities. Recent studies (Blau & Michaeli, 2018; 2019; Zhang et al., 2021) have extended this theory to include perceptual quality, revealing a three-way trade-off among rate, distortion, and perception. In this work, we leverage RD theory to characterize latent distortions arising from model mismatch. Our analysis reveals that forged latents exhibit measurable geometric drift, which aids in the separation of forged samples.

## 3. Preliminary

### 3.1. Diffusion Models and DDIM Inversion

Denoising Diffusion Probabilistic Models (DDPM) (Ho et al., 2020) formulate the process of adding and removing noise as a Markov chain. Denoising Diffusion Implicit

Models (DDIM) (Song et al., 2021) extend DDPM to generate high-quality images with fewer sampling steps. Unlike DDPM, DDIM follows a deterministic and non-Markovian process, which enables reversible noising and denoising. To reduce memory usage and computational cost, Latent Diffusion Models (LDM) (Rombach et al., 2022) perform the diffusion process in a latent space. Given an image $x \in \mathbb{R}^{H \times W \times 3}$, LDM employs an encoder $\mathcal{E}(\cdot)$ maps $x$ to its latent representation $z_0 = \mathcal{E}(x)$, and a decoder $\mathcal{D}(\cdot)$ reconstructs the image as $x' = \mathcal{D}(z_0)$. Let $\beta_t$ denote the variance schedule at timestep $t$, where $t \in \{0, 1, \ldots, T-1\}$, and define $\bar{\alpha}_t = \prod_{i=1}^{t} \alpha_i = \prod_{i=1}^{t}(1 - \beta_i)$. At each denoising step, a learned noise predictor $\epsilon_\theta(z_t, t, C)$ estimates the noise added to $z_0$. The corresponding estimate of $z_0$ at timestep $t$ is given by:

$$\hat{z}_0^t = \frac{z_t - \sqrt{1 - \bar{\alpha}_t}\, \epsilon_\theta(z_t, t, C)}{\sqrt{\bar{\alpha}_t}}, \tag{1}$$

where $C$ denotes the text condition. Using $\hat{z}_0^t$, the latent at the previous timestep can be computed as:

$$z_{t-1} = \sqrt{\bar{\alpha}_{t-1}}\, \hat{z}_0^t + \sqrt{1 - \bar{\alpha}_{t-1}}\, \epsilon_\theta(z_t, t, C). \tag{2}$$

Diffusion inversion reverses the generative process by recovering the latent representation from a given image. DDIM inversion accomplishes this by reversing the time steps and applying the same update rule used in DDIM generation. Starting from the latent representation $z_0$, noise is incrementally added, with the $t$-th step defined as:

$$z_{t+1} = \sqrt{\bar{\alpha}_{t+1}}\, \hat{z}_0^t + \sqrt{1 - \bar{\alpha}_{t+1}}\, \epsilon_\theta(z_t, t, C). \tag{3}$$

### 3.2. Black-Box Forgery Attacks

In the black-box forgery setting (Müller et al., 2025), an adversary is given a watermarked image $x^{(w)}$ generated by the SP's model $\Theta$ and has only black-box query access to this model. The adversary relies on a proxy model $\Theta_\mathcal{A}$ to invert $x^{(w)}$ and obtain an estimated latent $\tilde{z}_T^{(w)}$. From this latent, two forgery strategies emerge: *Guidance-* and *Optimization-based* methods. While guidance-based methods initialize the reverse diffusion process from $\tilde{z}_T^{(w)}$ using auxiliary textual or structural conditions, optimization-based methods seek a perturbation $\delta$ to align a perturbed cover latent $\tilde{z}_T^{(c)} + \delta$ with $\tilde{z}_T^{(w)}$ under the proxy inversion $\mathcal{I}_\mathcal{A}$, such that $\mathcal{I}_{0 \rightarrow T}(\tilde{z}_T^{(c)} + \delta; u_\mathcal{A}) \approx \tilde{z}_T^{(w)}$ (See Sec. A.2 for details).

### 3.3. Rate-Distortion Theory

The RD function (Koga et al., 2013) aims to study the relation between the input $X$ and the output $\hat{X}$ of an encoder-decoder pair, and is a mapping defined by a conditional distribution $p_{\hat{X}|X}$. For a Gaussian source $X \sim \mathcal{N}(0, \sigma^2)$ under common distortion measures (*e.g.*, mean-square error), the RD function admits a closed form:

**Definition 3.1** (RD Function of a Gaussian Source)**.** Let $X \sim \mathcal{N}(0, \sigma^2)$ be a Gaussian source. The RD function under mean-squared error distortion $D$ is defined as:

$$R(D) = \begin{cases} \dfrac{1}{2} \log\left(\dfrac{\sigma^2}{D}\right) & 0 < D < \sigma^2, \\ 0 & D > \sigma^2. \end{cases} \tag{4}$$

Since the latent variable $z_T$ in diffusion models follows an isotropic Gaussian distribution $\mathcal{N}(0, \mathbf{I})$, we adopt this form to characterize the RD trade-off under model mismatch (see Sec. A.3 for details).

## 4. Methodology and Theoretical Analysis

### 4.1. Threat Model

**Adversary's capability:** Let $\hat{z}_T^{(w)}$, $\hat{z}_T^{(f)}$ denote the recovered latents of watermarked and forged samples, respectively. The adversary's objective is to generate forged images that successfully deceive both the watermark detector and extractor by minimizing the difference between $z_T^{(w)}$ and $\hat{z}_T^{(w)}$. To achieve this, a proxy model $\Theta_\mathcal{A}$ invert a watermarked image $x^{(w)}$ and obtain $\hat{z}_T^{(w)}$, which is then decoded by $\Theta_\mathcal{A}$ to produce the forged image $\hat{x}$. Meanwhile, the adversary aims to preserve the visual fidelity of $\hat{x}$, ensuring the forgery remains indistinguishable from genuine.

**Adversary's knowledge:** We assume a black-box setting in which the adversary has limited knowledge of the semantic watermarking methods and the SP's model. Specifically, in practice, the adversary does not know the model architecture or parameters, nor does it have access to the prompts used by legitimate users. However, the adversary can acquire watermarked images that are publicly shared or uploaded by users and are known to originate from a generative model.

**Provider's capability:** We assume that the SP has full control over the generative pipeline (*e.g.*, watermark embedding or detection) and complete knowledge of the model architecture and its parameters. However, storing the original generative features for every synthesized image is unfeasible in large-scale deployments. Therefore, in our threat model, the provider must rely on the latent noise $\hat{z}_T$ obtained through an inversion process to detect forged samples.

### 4.2. Theoretical Framework

**Rate–Distortion Perspective.** We study the theoretical limits of semantic watermark forgery from an information-theoretic perspective inspired by RD theory. Under this view, the forward and reverse processes of generative models are abstracted as the encoding and decoding stages of a lossy compression, in which the adversary's reconstruction acts as a lossy decoder approximating a target watermarked latent.

Let $X$ denote the target latent source, $\hat{X}$ its lossy recon-

struction, and $Y_w$ the corresponding decoded watermarked image in the data space (see Fig. 2). This formulation serves as an abstraction for analyzing fundamental limits, rather than an exact equivalence to the diffusion inversion process.

**Assumptions.** Our analysis relies on two standard assumptions regarding latent statistics and the distortion measure. First, we assume that latent representations follow a Gaussian distribution with shared covariance (Assumption B.1). Second, we adopt the MSE as the distortion measure for the information-theoretic analysis (Assumption B.2). These assumptions facilitate a tractable characterization of fundamental distortion limits. We emphasize that the geometric metrics introduced in Sec 4.3 are not used as distortion measures in the rate–distortion sense, but rather to characterize how inevitable distortion manifests in the latent space. Further theoretical analysis is provided in Sec. B, with empirical validations in Sec. D.

### 4.2.1. RATE-DISTORTION LOWER BOUND UNDER MODEL MISMATCH

As illustrated in Fig. 2, an adversary must rely on a proxy model $\Theta_{\mathcal{A}}$ that generally differs from the target model $\Theta$. This mismatch imposes intrinsic limits on the achievable inversion accuracy. We characterize these constraints in terms of an irreducible information loss and a corresponding effective rate penalty, which together yield a lower bound on the achievable distortion.

**Definition 4.1** (Irreducible Information Loss and Effective Rate Penalty)**.** Let $P_\Theta(\hat{X} \mid Y_w)$ and $P_{\Theta_{\mathcal{A}}}(\hat{X} \mid Y_w)$ denote the target and proxy posterior distributions, respectively. The irreducible information loss induced by model mismatch is defined as

$$D_{\text{irr}} := \mathbb{E}_{Y_w}\left[D_{\text{KL}}\left(P_\Theta(\cdot \mid Y_w) \| P_{\Theta_{\mathcal{A}}}(\cdot \mid Y_w)\right)\right].$$

Under the Gaussian latent approximation with variance $\sigma^2$, we instantiate the corresponding effective rate penalty as

$$I_{\text{pen}} := \frac{1}{2} \log_2 \left(1 + \frac{D_{\text{irr}}}{\sigma^2}\right).$$

Here, $D_{\text{irr}}$ measures the posterior discrepancy caused by model mismatch, while $I_{\text{pen}}$ serves as a Gaussian surrogate for the reduction in the adversary's usable information rate.

**Theorem 4.2** (Rate-Distortion Lower Bound under Model Mismatch)**.** *Under Assumptions B.1 and B.2, the minimal achievable distortion for an adversary operating at information rate $R$, denoted $D_{\min}(R)$, is lower-bounded by*

$$D_{\min}(R) \geq D_{\text{irr}} + \sigma^2 \cdot 2^{-2(R-I_{\text{pen}})}. \quad (5)$$

*Remark* 4.3. The positivity of the distortion floor follows from the nonzero posterior mismatch. When $D_{\text{irr}} > 0$, the target and proxy posteriors are distinct. Pinsker's inequality

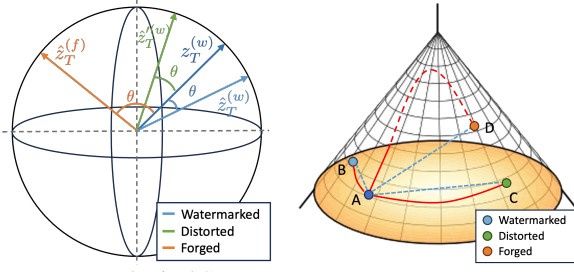

*(a)* Hyperspherical Space  *(b)* SPD Riemannian Manifold

*Figure 3.* Geometric interpretation of latent discrepancy. (a) illustrates angular separation $\theta$ on a hypersphere; (b) depicts the contrast between the geodesic distance (red solid line) and the Euclidean distance (blue dashed line) on the SPD manifold, with the dark blue dot denoting the watermarked latent $z_T^{(w)}$.

provides a standard control of their total-variation discrepancy in terms of the KL mismatch. Therefore, the effective penalty $I_{\text{pen}}$ in Def. 4.1 prevents the expected distortion from vanishing.

From Thm. 4.2, we identify two key consequences of model mismatch. First, the irreducible information loss $D_{\text{irr}}$ introduces a rate-independent distortion floor that cannot be eliminated by increasing the information rate $R$. Second, the penalty $I_{\text{pen}}$ reduces the adversary's usable information rate, leading to a slower decay of distortion as $R$ increases. Together, these effects indicate that high-fidelity semantic forgery is intrinsically constrained when the adversary relies on an imperfect proxy model.

The following corollary formalizes the existence of an irreducible distortion implied by Thm. 4.2.

**Corollary 4.4** (Existence of Distortion Error)**.** *Under the conditions of Theorem 4.2, there exists a constant $\epsilon > 0$ such that $D_{\min}(R) \geq \epsilon$ for any rate $R$. In particular, even with unbounded rate or computational resources, a black-box adversary cannot achieve arbitrarily small inversion distortion under model mismatch.*

These distortions can be attributed to both *intrinsic* inaccuracies of the target model and *external* errors induced by proxy models; a detailed description is deferred to Sec. C.1.

### 4.3. Geometric Interpretation of Latent Discrepancy

#### 4.3.1. FROM DISTORTION TO LATENT GEOMETRY

Under a Gaussian approximation, the latent space features a well-defined geometric structure, shared by both the recovered latent $\hat{z}_T$ and the reference latent $z_T$. In contrast, for forged samples, the distortion induced by a proxy model disrupts this intrinsic structure, resulting in systematic, measurable geometric deviations (see Fig. 3). Such a discrepancy enables the SP to distinguish forged samples by comparing a given recovered latent $\hat{z}_T$ with the reference watermarked

latent $z_T^{(w)}$ from the target model (see Fig. 8 in Appendix). In the following, we examine how such geometric distortion manifests along specific dimensions of the latent space.

### 4.3.2. DIRECTIONAL DRIFT ON THE HYPERSPHERE

With an isotropic Gaussian prior $\mathcal{N}(0, \mathbf{I}_N)$, where $N$ denotes the dimension of the latent space, any latent $\tilde{z}$ admits a polar decomposition $r \cdot u$, where the magnitude $r = \|\tilde{z}\|_2$ satisfies $r^2 \sim \chi^2(N)$ and the direction $u$ is uniformly distributed on the unit hypersphere $\mathbb{S}^{N-1}$. Since $r$ and $u$ are statistically independent, this decomposition effectively decouples the radial energy from the latent's geometric orientation. In high dimensions ($N \gg 1$), the radial component $r$ concentrates sharply around its mean, making magnitude variations negligible. Thus, magnitude variations become uninformative, and the discriminative geometric variation is almost entirely preserved in the angular component $u$.

This rotational invariance further implies that semantic perturbations from forgeries cannot be absorbed by radial variation and instead manifest as angular distortions, as shown in Fig. 3a. We treat such perturbations as geometric noise that drives the recovered latent $\hat{z}_T$ to deviate from the original $z_T$ on the hypersphere. To characterize this discrepancy, we define the *Spherical Angular Distortion* (SAD):

**Definition 4.5** (Spherical Angular Distortion (SAD)). Let $z_T, \hat{z}_T \in \mathbb{R}^N$ denote the original and recovered latent, respectively. The *Spherical Angular Distortion* (SAD) between $z_T$ and $\hat{z}_T$ is defined as the geodesic distance between their normalized projections on the unit hypersphere $\mathbb{S}^{N-1}$,

$$\mathrm{SAD}(z_T, \hat{z}_T) = \arccos\big(S_{\cos}(z_T, \hat{z}_T)\big),$$

where $S_{\cos}(z_T, \hat{z}_T) = \frac{\langle z_T, \hat{z}_T \rangle}{\|z_T\|_2 \|\hat{z}_T\|_2}$ denotes the cosine similarity between $z_T$ and $\hat{z}_T$.

**Lemma 4.6** (Unavoidable Angular Distortion under Forgery Perturbations). *Let $z_T^{(w)} \in \mathbb{R}^N \sim \mathcal{N}(0, \mathbf{I}_N)$. For a recovered watermarked latent $\hat{z}_T^{(w)}$, the orientation remains well aligned, such that $\mathrm{SAD}(z_T^{(w)}, \hat{z}_T^{(w)}) \approx 0$. Conversely, for a forged recovered latent $\hat{z}_T^{(f)} = z_T^{(w)} + \delta$ stemming from a mismatched proxy model $\Theta_{\mathcal{A}}$, the distortion satisfies: $\mathbb{E}\Big[\mathrm{SAD}(z_T^{(w)}, \hat{z}_T^{(f)})\Big] > 0$. In high-dimensional regimes ($N \gg 1$), this angular displacement remains bounded away from zero, acting as a manifest signature of forgery.*

Lem. 4.6 establishes that the orientation of the watermarked latent is inherently sensitive to proxy-induced noise. From a geometric perspective, this angular drift can be quantified through the alignment of latent vectors on $\mathbb{S}^{N-1}$.

*Remark* 4.7 (Implementation via Cosine Similarity). Since the geodesic (angular) distance $\theta = \arccos(S_{\cos})$ is a monotonically decreasing in $S_{\cos}$ on $[0, \pi]$, the angular distortion characterized in Lem. 4.6 can be equivalently captured by

$S_{\cos}$. Thus, we employ cosine similarity as a practical metric to evaluate global directional drift on the hypersphere.

### 4.3.3. DEFORMATION ON THE SPD MANIFOLD

Beyond point-wise drift, forgery perturbations compromise the local structural integrity of the latent manifold. As shown in Fig. 3b, we adopt a complementary perspective by mapping local statistics onto the manifold of Symmetric Positive Definite (SPD) matrices (Bhatia, 2009), $\mathcal{P}_N$. When endowed with the *Affine-Invariant Riemannian Metric* (AIRM) (Pennec et al., 2006), $\mathcal{P}_N$ forms a smooth manifold where the geodesic distance between $P_1, P_2 \in \mathcal{P}_N$ (the red solid path in Fig. 3b) is:

$$d_{\mathrm{AIRM}}(P_1, P_2) = \big\|\log(P_1^{-1/2} P_2 P_1^{-1/2})\big\|_F,$$

where $\|\cdot\|_F$ denotes the Frobenius norm. This metric is *congruence-invariant*, ensuring that the intrinsic geometry of covariance representations remains preserved across various distortion attacks. By contrast, the Euclidean distance (the blue dashed line in Fig. 3b) ignores the intrinsic curvature of the SPD manifold and lacks the invariance properties for detection.

**Definition 4.8** (Local SPD Geometric Inconsistency (LGI)). Let $\mathcal{B}(z_T) = \{t_i\}_{i=1}^M$ denote a local neighborhood of tokens around a latent $z$. We define the local covariance mapping $\mathcal{C} : z \mapsto P \in \mathcal{P}_N$ as

$$\mathcal{C}(z) := \frac{1}{M} \sum_{i=1}^M (t_i - \bar{t})(t_i - \bar{t})^\top + \varepsilon \mathbf{I}_N,$$

where $\bar{t}$ is the sample mean and $\varepsilon > 0$. The *Local SPD Geometric Inconsistency* is then defined using AIRM:

$$\mathrm{LGI}(z_T, \hat{z}_T) = d_{\mathrm{AIRM}}(\mathcal{C}(z_T), \mathcal{C}(\hat{z}_T)).$$

**Lemma 4.9** (Inevitable Structural Deformation under Forgery Perturbations). *Let $z_T^{(w)} \in \mathbb{R}^N$ be the initial watermarked latent with its associated mapping $\mathcal{C}(z_T^{(w)}) \in \mathcal{P}_N$. For a recovered watermarked latent $\hat{z}_T^{(w)}$, the local structural coherence is preserved such that $\mathrm{LGI}(z_T^{(w)}, \hat{z}_T^{(w)}) \approx 0$. Conversely, consider a forged recovered latent $\hat{z}_T^{(f)} = \hat{z}_T^{(w)} + \delta$ that induces a non-congruent deformation of the local token neighborhood $\mathcal{B}$. Under such structural perturbations, the resulting covariance $\mathcal{C}(\hat{z}_T^{(f)})$ is no longer related to $\mathcal{C}(z_T^{(w)})$ by a congruent transformation, and thus the induced inconsistency satisfies $\mathbb{E}\Big[\mathrm{LGI}(z_T^{(w)}, \hat{z}_T^{(f)})\Big] = \Omega(1)$.*

Instead of uniform scaling, forgery attacks induce non-uniform deformations that disrupt the relative positioning between neighboring tokens. By measuring distances on $\mathcal{P}_N$, we can distinguish between origin (target) and forgery-induced (proxy) recovered latent, as the latter exhibits anomalous geometric behavior.

*Table 1.* Watermark TPR under guidance-based forgery attacks (TPR@$X$-FPR).

| | | Frequency-domain | | | | Bitstream-level | |
|---|---|---|---|---|---|---|---|
| Proxy | Target | TR | RingID | HSTR | HSQR | GS | TAG |
| SD2.1 | SD2.1 | 0.991 | 1.000 | 0.996 | 0.993 | 0.999 | 1.000 |
| | SDXL | 0.976 | 1.000 | 0.991 | 0.989 | 0.999 | 0.999 |
| | PixArt-$\Sigma$ | 0.855 | 1.000 | 0.656 | 0.614 | 0.943 | 0.946 |
| | FLUX | 0.161 | 0.892 | 0.002 | 0.095 | 0.588 | 0.550 |
| | SD3 | 0.657 | 0.187 | 0.000 | 0.478 | 0.819 | 0.868 |
| SD3 | SD2.1 | 0.972 | 0.966 | 0.224 | 0.869 | 0.991 | 0.974 |
| | SDXL | 0.957 | 0.994 | 0.612 | 0.974 | 0.992 | 0.992 |
| | PixArt-$\Sigma$ | 0.887 | 0.994 | 0.507 | 0.968 | 0.943 | 0.958 |
| | FLUX | 0.612 | 0.582 | 0.022 | 0.889 | 0.995 | 0.981 |
| | SD3 | 0.849 | 0.836 | 0.343 | 0.973 | 0.996 | 0.997 |

*Table 2.* Our detection performance (AUC) under guidance-based attacks. "G-Cos" and "L-SPD" denote the global and local geometric deviation metric in Sec. 4, respectively.

| | | Frequency-domain | | | | | | | | Bitstream-level | | | |
|---|---|---|---|---|---|---|---|---|---|---|---|---|---|
| | | TR | | RingID | | HSTR | | HSQR | | GS | | TAG | |
| Proxy | Target | G-Cos | L-SPD | G-Cos | L-SPD | G-Cos | L-SPD | G-Cos | L-SPD | G-Cos | L-SPD | G-Cos | L-SPD |
| SD2.1 | SD2.1 | 0.955 | 0.929 | 0.982 | 0.968 | 0.983 | 0.966 | 0.958 | 0.933 | 0.975 | 0.948 | 0.974 | 0.964 |
| | SDXL | 1.000 | 0.997 | 1.000 | 0.998 | 1.000 | 0.997 | 1.000 | 0.998 | 1.000 | 0.995 | 1.000 | 0.988 |
| | PixArt-$\Sigma$ | 1.000 | 0.992 | 1.000 | 0.981 | 1.000 | 0.953 | 1.000 | 0.998 | 1.000 | 0.995 | 1.000 | 0.875 |
| | FLUX | 1.000 | 0.993 | 1.000 | 0.996 | 1.000 | 0.988 | 1.000 | 0.999 | 1.000 | 1.000 | 1.000 | 0.994 |
| | SD3 | 1.000 | 1.000 | 0.923 | 0.832 | 1.000 | 0.996 | 1.000 | 0.988 | 1.000 | 0.999 | 1.000 | 0.922 |
| SD3 | SD2.1 | 1.000 | 0.989 | 1.000 | 0.997 | 1.000 | 0.999 | 1.000 | 0.985 | 1.000 | 0.999 | 1.000 | 0.993 |
| | SDXL | 1.000 | 0.991 | 1.000 | 0.994 | 1.000 | 0.992 | 1.000 | 0.997 | 1.000 | 0.985 | 1.000 | 0.977 |
| | PixArt-$\Sigma$ | 1.000 | 0.990 | 1.000 | 0.984 | 1.000 | 0.927 | 1.000 | 0.994 | 1.000 | 0.988 | 1.000 | 0.847 |
| | FLUX | 0.997 | 0.995 | 0.999 | 0.994 | 0.911 | 0.880 | 0.997 | 0.998 | 0.998 | 1.000 | 0.999 | 0.993 |
| | SD3 | 0.943 | 1.000 | 0.919 | 0.832 | 0.981 | 0.982 | 0.939 | 0.919 | 0.933 | 0.993 | 0.944 | 0.804 |

### 4.3.4. GEOMETRIC IMPLICATIONS OF DISTORTION

The established lemmas cast forgery detection as a manifold separability problem. By projecting recovered latents into the joint bi-metric space, the inherent geometric displacement implies that forged samples are statistically distinguishable from watermarked ones, as formalized below:

**Theorem 4.10** (Probabilistic Geometric Separability under Black-Box Forgery). *Let $z_T^{(w)} \in \mathbb{R}^N$ be the initial watermarked latent, and let $\hat{z}_T^{(w)}$ and $\hat{z}_T^{(f)}$ denote the recovered latents corresponding to the watermarked and forged samples, respectively. Given the conditions in Lemma 4.6 and Lemma 4.9, there exists a rejection region $\mathcal{R} \subset \mathbb{R}_+^2$ such that the joint geometric distortion satisfies:*

$$\Pr\left[(\mathrm{SAD}(z_T^{(w)}, \hat{z}_T^{(f)}), \mathrm{LGI}(z_T^{(w)}, \hat{z}_T^{(f)})) \in \mathcal{R}\right] \geq 1 - e^{-\Omega(N)}.$$

*Conversely, for any watermarked recovered latent $\hat{z}_T^{(w)}$, the pair $(\mathrm{SAD}, \mathrm{LGI})$ remain concentrated outside $\mathcal{R}$ with high probability, as $(\mathrm{SAD}, \mathrm{LGI}) \to (0, 0)$ for $\hat{z}_T^{(w)}$. Thus, as $N \to \infty$, forged and watermarked latents become asymptotically separable in the joint geometric space.*

The proof of Thm. 4.10 can be found in Sec. B. Thm. 4.10 formalizes the complementary effects of global drift (Lem. 4.6) and local deformation (Lem. 4.9), ensuring robust detection even under partial perturbations. In practice, however, finite-dimensional overlap may persist if: (*i*) *High Proxy Fidelity* minimizes trajectory drift; or (*ii*) *Robustness Margins* allow heavily distorted watermarked samples to exhibit geometric features akin to those of forgeries. Therefore, perfect separability cannot be guaranteed in finite dimensions, despite high-probability separability.

## 5. Experiments

### 5.1. Experimental Settings

**Models and Datasets.** We consider two adversarial scenarios: guidance-based and optimization-based. For the guidance-based scenario, we focus on *reprompting* attacks

following (Müller et al., 2025), using Stable Diffusion 2.1 (SD2.1) (Rombach et al., 2022) and Stable Diffusion 3 (SD3) (Esser et al., 2024) as proxy models and five commonly used target models: SD2.1, SDXL (Podell et al., 2024), PixArt-$\Sigma$ (Chen et al., 2024), FLUX.1 (Labs, 2024), SD3. This evaluation is conducted on 1,000 samples. Regarding the optimization-based scenario, we randomly select 100 cover images from the MS-COCO dataset (Lin et al., 2014) and use SD2.1 as the proxy model with the same target set. All experiments generate images at a size of $512 \times 512$ using prompts from Stable-Diffusion-Prompt[1]. Further details are provided in Sec. D.

**Watermarking Methods.** We consider six state-of-the-art semantic watermarks: TR (Wen et al., 2023), RingID (Ci et al., 2024), HSTR (Lee & Cho, 2025), HSQR (Lee & Cho, 2025), GS (Yang et al., 2024b) and TAG (Chen et al., 2025). The first four methods embed watermarks in the Fourier frequency domain, while the latter two encode the bitstreams as Gaussian perturbations within the initial latent noise. We refer the reader to Sec. A for more details.

**Evaluation Metrics.** For watermark detection, we report the true positive rate (TPR) at fixed false positive rates (FPRs), using $10^{-3}$ for frequency-domain schemes and a stricter $10^{-6}$ for bitstream-level schemes. The corresponding detection thresholds are determined by $p$-values (for TR) and $\ell_1$ distance in frequency-domain methods; bitstream-level schemes are based on bit accuracy. Furthermore, we adopt the Area Under the Curve (AUC) as a threshold-independent metric to evaluate our detection performance.

**Our Detection Details.** To quantify latent geometric discrepancies, we use global cosine similarity (G-Cos) as an empirical realization of SAD to capture directional drift, and localized SPD-based distance (L-SPD) as a measure of LGI to characterize structural deformation. For L-SPD, we partition the latent space into a regular $16 \times 16$ spatial grid and compute SPD covariance matrices for each patch. We then aggregate these localized discrepancies using a top-$k$ mean strategy (with $k = 5$) to identify the most significant regional distortions.

---

[1]Stable-Diffusion-Prompts

*Table 3.* Watermark TPR under optimization-based forgery attacks (TPR@$X$-FPR).

| Target | Step | Frequency-domain | | | | Bitstream-level | |
|---|---|---|---|---|---|---|---|
| | | TR | RID | HSTR | HSQR | GS | TAG |
| SD2.1 | 20 | 0.95 | 0.98 | 0.67 | 0.99 | 1.00 | 1.00 |
| | 50 | 0.99 | 1.00 | 0.99 | 1.00 | 1.00 | 1.00 |
| | 100 | 1.00 | 1.00 | 1.00 | 1.00 | 1.00 | 1.00 |
| SDXL | 20 | 0.26 | 0.35 | 0.04 | 0.30 | 0.97 | 0.96 |
| | 50 | 0.67 | 0.85 | 0.20 | 0.76 | 0.99 | 0.99 |
| | 100 | 0.83 | 1.00 | 0.55 | 0.94 | 1.00 | 0.99 |
| PixArt-Σ | 20 | 0.18 | 0.21 | 0.01 | 0.02 | 0.43 | 0.33 |
| | 50 | 0.42 | 0.59 | 0.03 | 0.20 | 0.94 | 0.92 |
| | 100 | 0.73 | 0.85 | 0.14 | 0.42 | 0.97 | 0.99 |
| FLUX.1 | 20 | 0.04 | 0.01 | 0.01 | 0.01 | 0.00 | 0.11 |
| | 50 | 0.13 | 0.02 | 0.01 | 0.01 | 0.43 | 0.60 |
| | 100 | 0.14 | 0.01 | 0.00 | 0.01 | 0.88 | 0.79 |
| SD3 | 20 | 0.18 | 0.00 | 0.00 | 0.00 | 0.20 | 0.28 |
| | 50 | 0.45 | 0.00 | 0.00 | 0.01 | 0.74 | 0.74 |
| | 100 | 0.63 | 0.00 | 0.00 | 0.04 | 0.94 | 0.89 |

*Table 4.* Our detection performance (AUC) under optimization-based attacks. "G-Cos" and "L-SPD" are used here as in the guidance-based attack scenarios (see Tab. 2).

| Target | Step | Frequency-domain | | | | | | | | Bitstream-level | | | |
|---|---|---|---|---|---|---|---|---|---|---|---|---|---|
| | | TR | | RID | | HSTR | | HSQR | | GS | | TAG | |
| | | G-Cos | L-SPD | G-Cos | L-SPD | G-Cos | L-SPD | G-Cos | L-SPD | G-Cos | L-SPD | G-Cos | L-SPD |
| SD2.1 | 20 | 1.000 | 0.978 | 1.000 | 0.999 | 1.000 | 0.999 | 1.000 | 0.979 | 1.000 | 0.998 | 1.000 | 0.997 |
| | 50 | 0.997 | 0.955 | 1.000 | 0.991 | 1.000 | 0.990 | 0.996 | 0.953 | 0.998 | 0.992 | 1.000 | 0.989 |
| | 100 | 0.978 | 0.905 | 0.999 | 0.958 | 0.999 | 0.971 | 0.968 | 0.911 | 0.990 | 0.970 | 0.994 | 0.961 |
| SDXL | 20 | 1.000 | 0.997 | 1.000 | 0.999 | 1.000 | 0.996 | 1.000 | 1.000 | 1.000 | 0.997 | 1.000 | 0.994 |
| | 50 | 1.000 | 0.997 | 1.000 | 0.997 | 1.000 | 0.993 | 1.000 | 0.999 | 1.000 | 0.995 | 1.000 | 0.986 |
| | 100 | 1.000 | 0.990 | 1.000 | 0.996 | 1.000 | 0.989 | 1.000 | 0.998 | 1.000 | 0.992 | 1.000 | 0.978 |
| PixArt-Σ | 20 | 1.000 | 0.956 | 1.000 | 0.999 | 1.000 | 0.995 | 1.000 | 1.000 | 1.000 | 0.957 | 1.000 | 0.985 |
| | 50 | 1.000 | 0.952 | 1.000 | 0.998 | 1.000 | 0.995 | 1.000 | 1.000 | 1.000 | 0.954 | 1.000 | 0.983 |
| | 100 | 1.000 | 0.954 | 1.000 | 0.995 | 1.000 | 0.992 | 1.000 | 1.000 | 1.000 | 0.950 | 1.000 | 0.976 |
| FLUX.1 | 20 | 1.000 | 0.994 | 1.000 | 0.993 | 1.000 | 0.991 | 1.000 | 1.000 | 1.000 | 1.000 | 1.000 | 1.000 |
| | 50 | 1.000 | 0.994 | 1.000 | 0.992 | 1.000 | 0.992 | 1.000 | 1.000 | 1.000 | 1.000 | 1.000 | 0.999 |
| | 100 | 1.000 | 0.995 | 1.000 | 0.991 | 1.000 | 0.992 | 1.000 | 1.000 | 1.000 | 1.000 | 1.000 | 0.999 |
| SD3 | 20 | 1.000 | 0.994 | 1.000 | 0.929 | 1.000 | 0.995 | 1.000 | 0.995 | 1.000 | 0.993 | 1.000 | 0.987 |
| | 50 | 1.000 | 0.994 | 1.000 | 0.931 | 1.000 | 0.992 | 1.000 | 0.995 | 1.000 | 0.993 | 1.000 | 0.987 |
| | 100 | 1.000 | 0.993 | 1.000 | 0.929 | 1.000 | 0.992 | 1.000 | 0.994 | 1.000 | 0.993 | 1.000 | 0.987 |

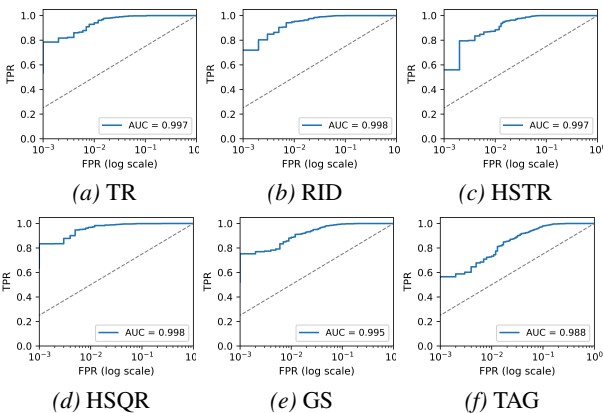

*(a)* TR     *(b)* RID     *(c)* HSTR

*(d)* HSQR     *(e)* GS     *(f)* TAG

*Figure 4.* Log-scale ROC curves for our local structural metric across watermarking schemes, evaluated on SDXL as the target model and SD2.1 as the proxy under guidance-based attacks.

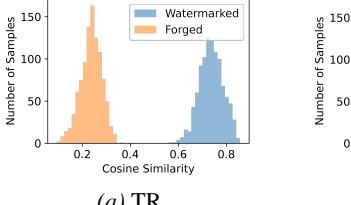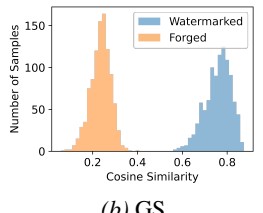

*(a)* TR      *(b)* GS

*Figure 5.* Distributions of cosine similarity for watermarked and forged samples, under the same attack setting as in Fig. 4.

## 5.2. Experimental Results

**Guidance-based Forgery Attacks.** Tab. 1 illustrates the robustness of various watermarking schemes under guidance-based attacks. We find that forgery attacks are highly successful when the target and proxy models are identical (*e.g.*, both are SD2.1); however, the attack efficacy is notably constrained in cross-model scenarios (*e.g.*, an SD3 target with an SD2.1 proxy), consistent with the analysis in Sec. 4.2.

Tab. 2 reports our detection performance measured by global and local geometric deviations. Both detection metrics, G-Cos and L-SPD, achieve near-perfect AUC values, exceeding 0.990 in many cases, reflecting substantial semantic deviations induced by architectural discrepancies between target and proxy models. By contrast, detection performance degrades slightly in the most challenging scenarios where target and proxy models are identical (*e.g.*, both are SD3) and approximation errors are minimized. Despite this, our method retains meaningful detection performance under these conditions (*e.g.*, above 0.990 in SD2.1). Figs. 4 and 5 present the detection performance using ROC curves and the semantic shift computed by cosine similarity, respectively.

**Optimization-based Forgery Attacks.** Tab. 3 reports the forgery success rates of watermarking schemes under optimization-based attacks, where higher values indicate greater attack effectiveness. When the target model matches the proxy (SD2.1), the adversary achieves near-perfect forgery success (*i.e.*, 1.00 TPR) even under limited steps. Conversely, for DiT-based target models (*e.g.*, FLUX.1 and SD3), the attack performance degrades significantly across all watermarking schemes.

Tab. 4 presents our detection performance under diverse conditions, following the same metrics as in Tab. 2. While G-Cos and L-SPD show a marginal decrease with more optimization steps, both retain strong detection capability, *e.g.*, the L-SPD metric decreases slightly from 0.997 to 0.990 for TR on SDXL. A similar trend is also evident in Fig. 6, which shows a clear separation between cosine similarity distributions at different optimization steps. Notably, a more significant performance drop is observed when the target model matches the proxy (*i.e.*, SD2.1), as discussed in Sec. 4.2. However, we argue that such a matched-model scenario is less representative of real-world threats.

**Comparison with Alternative Metrics.** Although our framework is not tied to a specific metric, G-Cos and L-SPD are chosen as empirical realizations of the geometric deviations characterized in Lemmas 4.6 and 4.9. We fur-

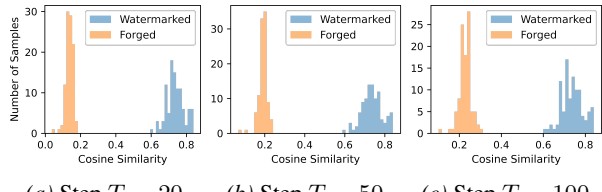

*(a)* Step $T = 20$   *(b)* Step $T = 50$   *(c)* Step $T = 100$

*Figure 6.* Cosine similarity distributions under optimization-based attacks, with increasing numbers of optimization iterations on the TR. The target and proxy models are consistent with Fig. 4.

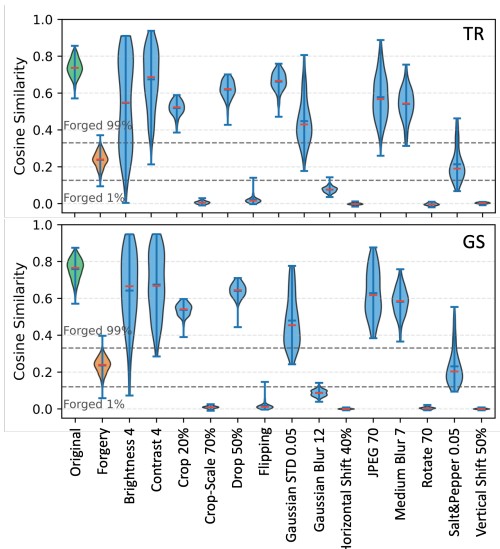

*Figure 7.* Similarity distributions under 14 types of image distortions for TR and GS, evaluated on SDXL as the target model.

ther compare against a local cosine baseline (*i.e.*, grid size 16, mean aggregation), which also achieves competitive performance in both cross-model and identical-model settings (*e.g.*, AUC $= 1.000$ for both TR and GS under SDXL $\rightarrow$ SD2.1, and $0.954/0.976$ in the identical-model setting). These results suggest that capturing geometric discrepancies is central to detection, while G-Cos and L-SPD provide principled geometric metrics.

### 5.3. Robustness Against Image Distortion Attacks

Fig. 7 evaluates the robustness of our method against various post-processing distortions. For each distortion and intensity level, we report the average performance over 100 randomly selected samples. The results indicate that our method maintains high geometric consistency under most distortions across both TR and GS. Nevertheless, in a few cases involving stochastic noise and nonlinear signal degradations, the similarity distributions exhibit increased dispersion. Such distortions partially corrupt the latent structural coherence, degrading the intrinsic watermarked features and verification performance (see Sec. D for details).

*Table 5.* Distortion between watermarked and reconstructed latents. Baseline error (*w/o*) represents the intrinsic reconstruction error without proxy guidance, while values in parentheses denote the empirical $D_{\mathrm{irr}}$ estimate.

| | TR | | | GS | | |
|---|---|---|---|---|---|---|
| Target | *w/o* | SD2.1$_{\mathcal{A}}$ ($D_{\mathrm{irr}}$) | SD3$_{\mathcal{A}}$ ($D_{\mathrm{irr}}$) | *w/o* | SD2.1$_{\mathcal{A}}$ ($D_{\mathrm{irr}}$) | SD3$_{\mathcal{A}}$ ($D_{\mathrm{irr}}$) |
| SD2.1 | 0.253 | 0.621 (0.368) | 1.285 (1.032) | 0.217 | 0.534 (0.317) | 1.262 (1.045) |
| SDXL | 0.405 | 1.222 (0.817) | 1.229 (0.824) | 0.430 | 1.239 (0.809) | 1.251 (0.821) |
| PixArt-$\Sigma$ | 0.473 | 1.244 (0.771) | 1.257 (0.784) | 0.480 | 1.252 (0.772) | 1.271 (0.791) |
| FLUX | 0.422 | 1.528 (1.106) | 1.322 (0.900) | 0.399 | 1.552 (1.153) | 1.333 (0.934) |
| SD3 | 0.573 | 1.496 (0.923) | 1.005 (0.432) | 0.570 | 1.512 (0.942) | 1.004 (0.434) |

### 5.4. Reconstruction Distortion Analysis

To empirically quantify the additional reconstruction error associated with $D_{\mathrm{irr}}$, we compute the MSE between $z_T$ and $\hat{z}_T$, as reported in Tab. 5. We compare $\mathrm{MSE}_{\mathrm{target}}$, obtained from the target model's standard inversion for watermark verification, with $\mathrm{MSE}_{\mathrm{proxy}}$, obtained after proxy-based inversion. We estimate the proxy-induced distortion as $\hat{D}_{\mathrm{irr}} = \mathrm{MSE}_{\mathrm{proxy}} - \mathrm{MSE}_{\mathrm{target}}$. For simplicity, the values in parentheses in Tab. 5 are reported as $D_{\mathrm{irr}}$, which denotes this empirical estimate.

When the target and proxy models share the same architecture (*e.g.*, SD2.1), both the reconstruction error and the estimated $\hat{D}_{\mathrm{irr}}$ remain small. For example, under TR with SD2.1 as both target and proxy, the MSE is $0.621$, with $\hat{D}_{\mathrm{irr}} = 0.368$. In contrast, cross-model settings produce larger distortion, reaching an MSE of $1.496$ for TR when targeting SD3 with an SD2.1 proxy. This shows that proxy-induced distortion is measurable and increases with target-proxy discrepancy, consistent with our formulation, although MSE alone is not reliable for detection under common image distortions (see Sec. D).

## 6. Discussions

### 6.1. Pseudo-randomness Property

PRC watermark (PRCW) (Gunn et al., 2025) incorporates pseudo-random error-correcting codes (PRC) (Christ & Gunn, 2024) to ensure that an adversary cannot distinguish between watermarked and unwatermarked images, even under adaptive query access. As shown in Tab. 9, this pseudo-randomness significantly enhances robustness against forgery attacks. However, successful forgery can still occur in the special case where the proxy and target models are identical. In this setting, our approach serves as an orthogonal defense that remains effective when model mismatch no longer provides protection. We provide further descriptions in Sec. D.

### 6.2. Impact of Proxy Model Architecture on Forgery

In Tab. 1, we observe that the proxy model with higher latent channel dimension (*i.e.*, SD3 with 16 channels) exhibits stronger forgery capability compared to those with lower di-

mensions (*i.e.*, SD2.1). For example, when SD2.1 serves as the target model, SD3 achieves a remarkably high detection rate (*e.g.*, $0.972$ for TR and $0.991$ for GS). This indicates that increased channel capacity enables the proxy to capture and manipulate finer-grained latent structures, facilitating more accurate alignment with the watermarked latent during forgery. Nevertheless, the unavoidable proxy–target mismatch still induces a structured geometric deviation in latent space, which our method consistently exploits to distinguish forged samples from watermarked ones (see Tab. 2).

### 6.3. Practical Limitations and Trade-offs

While capturing robust local descriptors (*e.g.*, SIFT (Lowe, 2004) or ORB (Rublee et al., 2011)) or deep feature correspondences (*e.g.*, DIFT (Tang et al., 2023)) at intermediate denoising or sampling steps could enhance forgery detection, such approaches incur prohibitive storage overhead for SPs. This requirement contradicts the zero-storage principles inherent in watermarking. Accordingly, our framework operates exclusively on the latent noise $z_T$ and $\hat{z}_T$. This design ensures that our approach remains lightweight and maintains seamless compatibility with existing schemes.

### 6.4. Security Boundaries under Stronger Attackers

The identical target-proxy setting represents an important security boundary in which the proxy-induced architectural mismatch ($\epsilon_{\mathrm{mis}}$) is minimized. However, the practical forgery boundary is constrained by an unavoidable internal accumulated error ($\epsilon_{\mathrm{int}}$), as discussed in Sec. C.1. This error arises from the asymmetry between the forward generation and inversion processes and accumulates throughout the diffusion trajectory, even under identical architectures. Consequently, exact trajectory reversal remains difficult in practice, leading to measurable latent discrepancies.

Empirically, even under this challenging attacker setting, our method achieves robust detection performance, with AUCs above $0.955$ for G-Cos and $0.928$ for L-SPD. In addition, the proposed pre-verification stage serves as an early warning mechanism by flagging samples with anomalous geometric drift as potential forgeries, even when watermark verification succeeds.

## 7. Conclusion

In this work, we demonstrate that the intrinsic limitations of black-box forgery attacks stem from proxy-target model mismatch, as formalized through the lens of rate–distortion theory. We reveal that this mismatch manifests as structured geometric deviations on the intrinsic latent manifold, characterized by global drift and local deformation. These geometric deviations provide a principled basis for identifying forged samples. Extensive experimental results validate the effectiveness of our approach across diverse black-box forgery scenarios. Our findings offer a new perspective for the design of more robust semantic watermarking schemes.

## Acknowledgements

This work was supported by the National Science and Technology Council (NSTC) with Grants NSTC 114-2221-E-001-010-MY2, 114-2634-F-001-001-MBK, 114-2634-F-002-004, and AS-IAIA-114-M10 for Academia Sinica.

## Impact Statement

This paper enhances the security of generative models by identifying forged watermarks through the lens of geometric distortion. Our approach helps mitigate the threats posed by black-box forgery attacks, thereby preventing unauthorized use and malicious dissemination. Overall, this work contributes to ongoing efforts toward the responsible and secure deployment of generative AI systems.

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

The content of Supplementary Material is summarized as follows: 1) In Sec. A, we provide the background information to facilitate a better understanding of our methodology; 2) In Sec. B, we state the underlying assumptions and provide formal proofs for Theorem 4.2 and Theorem 4.10; 3) In Sec. C, we offer an in-depth analytical extension of our main findings, focusing on the theoretical underpinnings of detection errors and the practical scope of our evaluation; 4) In Sec. D, we elaborate on our implementation, including the datasets, the model architectures, and an extended set of experimental evaluations; 5) Finally, Sec. E showcases the visual examples of forged images generated by guidance- and optimization-based black-box forgery attacks.

## A. Background

### A.1. Semantic Watermarking Methods

**Tree-Ring (TR).** Tree-Ring (Wen et al., 2023) introduces the concept of inversion-based watermarking, which can be divided into two phases: ($i$) *Generation* and ($ii$) *Detection*.

● *Generation.* During image generation, TR injects a concentric circular pattern into the frequency representation of a clean initial latent $z_T \sim \mathcal{N}(0, \mathbf{I}_N)$, generating a watermarked noise $z_T^{(w)}$. This latent then serves as the starting point for the diffusion process, which involves denoising and decoding to generate the final watermarked image $x^{(w)}$.

● *Detection.* We first reconstruct the initial noise $\hat{z}_T^{(w)} = \mathcal{I}_{0 \to T}(\mathcal{E}(x^{(w)}); \mathcal{U})$ and analyze its frequency spectrum. A statistical test aggregates the squared absolute differences between observed and expected frequency values across all rings; a resulting $p$-value below the threshold $\tau$ indicates the presence of the watermarked pattern.

In this paper, we use a ring pattern with a radius of 10 and apply zero-bit watermarking. Following the prior work (Müller et al., 2025), we adopt the same detection thresholds derived from statistics on 5,000 watermarked and 5,000 clean images to achieve the target false positive rate (FPR).

**RingID.** As a subsequent advancement of TR, RingID (Ci et al., 2024), ensures that the distribution of $z_T^{(w)}$ more closely aligns with $\mathcal{N}(0, \mathbf{I}_N)$. Furthermore, it introduces support for multi-bit watermarking, enabling the embedding of multi-bit messages for user identification. Note that we follow the default parameters of RingID for implementation.

**Symmetric Fourier-based Watermarks.** Compared to existing frequency-domain methods, Symmetric Fourier-based watermarks (Lee & Cho, 2025) ensure frequency integrity by enforcing Hermitian symmetry (HS). Based on this principle, the authors introduce two semantic variants, HSTR and HSQR, both of which exhibit superior performance and robustness over established baselines such as TR and RingID. To ensure a fair comparison, our experimental settings align with the original (Lee & Cho, 2025) (*i.e.*, central-aware embedding with a $44 \times 44$ frequency mask, and HSQR configured with QR version 1 and a box size of 2).

**Gaussian Shading (GS).** Gaussian Shading (Yang et al., 2024b) is the first bit-stream level semantic watermarking by incorporating the cryptographic primitive to achieve preserving generation. We describe the two phases of GS as follows:

● *Generation.* Given a message $s$ of length $k$, we first replicate it $\rho$ times to obtain $s^d$, which is then encrypted using the symmetric stream cipher ChaCha20 (Bernstein, 2008). The cipher takes a secret key and a unique random seed as input for each image, generating an encrypted bitstream $m$. The encrypted message $m$ is then used to modulate the sampling process of the initial latent $z_T^{(w)}$. Specifically, GS partitions the Gaussian distribution into $2^\ell$ bins with equal probability. In this work, we adopt the standard setting with $\ell = 1$, which effectively bisects the Gaussian distribution into two regions corresponding to negative and positive values. Each corresponding latent bit of $m[i] \in \{0, 1\}$ determines whether $z_T^{(w)}[i]$ is sampled from the negative or the positive region of the Gaussian distribution. In addition, the encryption ensures that the bits in $m$ are uniformly distributed, thereby preserving the Gaussian distribution of $z_T^{(w)}$. The watermarked image is then generated by continuing the standard sampling process: $x^{(w)} = \mathcal{G}_{T \to 0}(z_T^{(w)}; \mathcal{U})$.

● *Verification.* To verify the watermark, GS first applies the inversion process $\mathcal{I}_{0 \to T}$ to obtain the estimated $\hat{x}_T^{(w)}$. The inverted latent $\hat{x}_T^{(w)}$ is then quantized to retrieve the encrypted message bits $m'$. Specifically, we set $m'[i] = 0$ if $\hat{z}_T^{(w)}[i] < 0$ and $m'[i] = 1$ otherwise. The recovered bitstream $m'$ is then decrypted to reconstruct the repeated message $s'^d$. Finally, the original message of $s'$ is determined via majority voting over $\rho$ replicated bits, which corrects bit errors and improves robustness to noise.

Following the settings in (Müller et al., 2025), we use an encoding window of $\ell = 1$, with a unique random key and message per image. The message length $k$ is 256, resulting in 1024 bits. The repetition factor $\rho$ is 64 for SD2.1, and 256 for FLUX.1 and SD3, which uses 16-channel latents compared to four channels in the other models. In the detection scenario, we count a true positive if $r(s, s')$ exceeds a threshold calibrated to achieve a specified FPR (*i.e.*, $10^{-6}$). The threshold is 0.70703. In addition, since our objective is centered on detection, we exclude attribution scenarios from this study.

In addition, Videoshield (Hu et al., 2025) adapts the GS scheme to both text-to-video and image-to-video diffusion models by incorporating intrinsic tamper localization. This training-free approach preserves spatio-temporal integrity while ensuring the latent trajectory remains consistent throughout the temporal domain. Crucially, our detection framework can be extended to assess the authenticity of each recovered latent frame, effectively distinguishing between watermarked recovered latents and forged recovered latents before watermark extraction. The extension of our method to such video-based scenarios is deferred to future work.

**TAG Watermarks.** Building on GS, TAG watermark (Chen et al., 2025) employs a dual-mark joint sampling strategy to co-embed copyright and localization watermarks without quality loss. By leveraging the sensitivity of the inversion process, it can localize manipulated regions to guide tamper-aware decoding, which excludes compromised bits and robustly restores messages under tampering. For our experiments, we embed the 256-bit watermark into the initial latent noise $z_T$ using the TAG scheme. To maximize error resilience, we implement an adaptive repetition strategy, where the redundancy factor $N_{rep}$ is dynamically determined by $N_{rep} = \lfloor \frac{\text{Dim}(z_T)}{256} \rfloor$, thereby ensuring the watermark payload fully populates the available latent dimensions $\text{Dim}(z_T)$. The threshold follows the same setting as GS.

**PRC Watermarks.** The PRC Watermark (PRCW) (Gunn et al., 2025) embeds information into the latent representation of diffusion models by using pseudorandom error-correcting codes (PRC) (Christ & Gunn, 2024). Instead of stream cipher techniques, PRCW focuses on achieving statistical undetectability alongside robustness, ensuring that an adversary cannot distinguish between watermarked and unwatermarked images, even after making many adaptive queries. However, recent studies (Lee et al., 2025; Wang et al., 2025) have exposed emerging threats in such schemes, ranging from direct signal removal to cryptanalytic exploits against the PRC mechanism. In our implementation, following the original settings in (Gunn et al., 2025), we set the undetectable parameter $t = 3$ and maintain a FPR of $10^{-5}$.

**Other Related Works.** Beyond the schemes discussed above, several studies (Li et al., 2025a;b) have proposed alternative approaches to enhance the consistency of semantic watermarks. GaussMarker (Li et al., 2025a) embeds watermark signals in both spatial and frequency domains, forming a dual-domain representation. To handle geometric transformations, it employs a learnable Gaussian Noise Restorer (GNR) that maps distorted latents back to a Gaussian distribution, mitigating the effects of rotation and cropping. Similarly, Shallow Diffuse (Li et al., 2025b) adopts a decoupling strategy that exploits the low-dimensional subspace of the diffusion process. By embedding the watermark primarily in the null space of this subspace, the method separates the watermark signal from the sampling trajectory. In practice, GaussMarker can be viewed as a combination of TR and GS, while Shallow Diffuse is derived from the TR scheme. Accordingly, in this work, we focus our comparisons on representative frequency-domain and bitstream-level watermarking schemes.

## A.2. Black-box forgery Attacks

**Guidance-based Methods.** An adversary performs guidance-based forgery (Müller et al., 2025; Zhu et al., 2025) by first applying DDIM inversion with a proxy model $\Theta_{\mathcal{A}}$ to estimate a latent noise vector $\hat{z}_T^{(w)}$ from the public watermarked image $x^{(w)}$. This estimated latent serves as the initialization for a guided reverse diffusion process. Guidance can be introduced either through a textual prompt $t$ or via a controllable model such as ControlNet (Zhang et al., 2023), which conditions generation on a cover image $x^{(c)}$.

In the latter case, a trainable control module $\mathcal{F}_{\mathcal{A}}$ extracts structural information from $x^{(c)}$ (*i.e.*, edge or depth representations) and an encoder $\mathcal{E}_{\mathcal{A}}$ maps this information into a visual condition embedding (Zhu et al., 2025). This embedding guides the generation process jointly with a textual embedding $\mathcal{T}_{\mathcal{A}}(t^{(c)})$, which is derived from a descriptive prompt $t^{(c)}$ associated with the cover image and conditioned on a pretrained, frozen U-Net ($\mathcal{U}_{\mathcal{A}}$).

We denote the resulting guided reverse diffusion process by the operator $\mathcal{G}$, which is parameterized by the U-Net and the control module,

$$\hat{z}'_0 = \mathcal{G}_{\mathcal{A}, T \to 0}(\hat{z}_T^{(w)} \mid \mathcal{E}_{\mathcal{A}}(x^{(c)}), \mathcal{T}_{\mathcal{A}}(t^{(c)}); \mathcal{U}_{\mathcal{A}}, \mathcal{F}_{\mathcal{A}}),$$

where $\mathcal{E}_{\mathcal{A}}(x^{(c)})$ and $\mathcal{T}_{\mathcal{A}}(t^{(c)})$ provide the visual and textual conditions, respectively. Finally, the decoder $\mathcal{D}$ maps the refined

latent vector $\hat{z}_0'$ back to the pixel space, producing the forged image $\hat{x}^{(w)}$.

**Optimization-based Methods.** In contrast to guidance-based methods that manipulate the reverse diffusion process, optimization-based forgery operates by directly solving for an optimal latent variable. The core objective is to find a minimal perturbation to a clean image's latent state which, upon forward diffusion to timestep $T$, aligns with the known latent representation of a target watermarked image. As demonstrated in (Müller et al., 2025), performing this optimization in the near-noiseless latent space at $t = 0$ is an effective strategy.

The process begins by encoding the cover image $x^{(c)}$ to obtain its latent representation $\hat{z}_0^{(c)} = \mathcal{E}_\mathcal{A}(x^{(c)})$. The optimization objective is then formulated as minimizing the squared $L_2$ distance between the diffused perturbed latent and the target, as defined by the loss function:

$$L_{\text{forgery}}(\delta) = \left\| \mathcal{I}_{0 \to T}(\hat{z}_0^{(c)} + \delta; u_\mathcal{A}) - \hat{z}_T^{(w)} \right\|_2,$$

where $\mathcal{I}_{0 \to T}$ denotes the deterministic forward diffusion process that applies noise according to the predefined schedule up to timestep $T$, and $u$ represents the denoising backbone model. The adversary applies gradient descent for up to $N$ steps to minimize this loss *w.r.t.* $\delta$. Once the optimal perturbation $\delta^*$ is found, the forged latent $\hat{z}_0^{(c)} + \delta^*$ is decoded using the proxy model to generate the final forged image $\hat{x}^{(c)}$.

## A.3. Rate-distortion Theory

Rate-distortion theory (Shannon et al., 1959; Koga et al., 2013) characterizes the fundamental trade-off between the rate used to represent samples from a data source $X \sim p_X$ and the expected distortion incurred in decoding those samples from their compressed representations. Formally, the relation between the input $X$ and output $\hat{X}$ of an encoder-decoder pair is a (possibly stochastic) mapping defined by some conditional distribution $p_{\hat{X}|X}$. The expected distortion is given by

$$\mathbb{E}[\Delta(X, \hat{X})], \tag{6}$$

where the expectation is over the joint distribution $p_{X,\hat{X}} = p_{\hat{X}|X} p_X$, and $\Delta : \mathcal{X} \times \hat{\mathcal{X}} \to \mathbb{R}^+$ is any full-reference distortion measure such that $\Delta(X, \hat{X}) = 0$ if and only if $X = \hat{X}$.

A fundamental result states that for an i.i.d. source $X$, the minimum achievable rate under a distortion constraint $D$ is given by the rate-distortion function:

$$R(D) = \min_{p_{\hat{X}|X}} I(X, \hat{X}) \quad \text{s.t.} \quad \mathbb{E}[\Delta(X, \hat{X})] \leq D, \tag{7}$$

where $I$ denotes mutual information (Cover & Thomas, 2012). Closed-form expressions for the rate-distortion function $R(D)$ exist only for a few source distributions and simple distortion measures (*e.g.*, mean-squared error or Hamming distance), but in general, $R(D)$ is known to be non-increasing, convex, and continuous.

## A.4. Symmetric Positive-Definite (SPD) Manifold and AIRM

The space of SPD matrices forms a differentiable manifold known as the SPD manifold, which has been successfully applied in various fields (Huang & Van Gool, 2017; Brooks et al., 2019; You & Park, 2021). Since SPD matrices lie on a curved, non-Euclidean space, standard similarity measures (*e.g.*, Euclidean distance or Pearson correlation) fail to capture their intrinsic geometry. To account for the manifold structure, we adopt the Affine-Invariant Riemannian Metric (AIRM) (Pennec et al., 2006) to measure the distance between two SPD matrices $P_1$ and $P_2$. The AIRM distance is defined as:

$$d(P_1, P_2) = \sqrt{\sum_{i=1}^{N} \ln^2 \lambda_i}, \tag{8}$$

where $\lambda_i$ are the eigenvalues of $P_1^{-1} P_2$. The AIRM is characterized by its invariance under affine transformations and inversion. This metric provides a principled framework for analyzing manifold-valued data and effectively mitigates the "*swelling effect*," a common artifact where the determinant of an intermediate matrix exceeds that of its endpoints in Euclidean space.

# B. Proofs of Theorems and Lemmas in Sec. 4

## B.1. Assumptions

**Assumption B.1** (Gaussian Approximation). We assume the adversary's inversion task can be modeled by conditional probability distributions that are approximately multivariate Gaussian. Specifically, the true posterior distribution (from the target model, $\Theta$) and the adversary's assumed posterior (from the proxy model, $\Theta_{\mathcal{A}}$) are given by:

$$P_{\Theta}(X_0|Y_w) \approx \mathcal{N}(\mu_{\Theta}(Y_w), \sigma^2 I_d), \quad P_{\Theta_{\mathcal{A}}}(X_0|Y_w) \approx \mathcal{N}(\mu_{\mathcal{A}}(Y_w), \sigma^2 I_d),$$

where $Y_w$ is the observed watermarked output, and $\sigma^2$ is a shared noise variance. The means $\mu_{\Theta}$ and $\mu_{\mathcal{A}}$ differ due to the model mismatch.

The shared-variance assumption in Assumption B.1 reflects a simplified setting in which the adversary has access to a generative model with similar uncertainty behavior. While variance may differ across architectures in practice, adopting a shared variance allows us to isolate the effect of posterior mean deviation, which constitutes the dominant source of mismatch in latent representations. Moreover, this assumption establishes a conservative analytical baseline: any additional variance mismatch would further increase the reconstruction distortion, reinforcing the theoretical limits derived in Sec. 4.

To further justify the shared-variance premise, we consider the underlying score-matching objective. According to Tweedie's Formula (Efron, 2011; Luo, 2022), the posterior mean of clean data $x_0$ is determined by the score function:

$$\mathbb{E}[x_0 \mid x_t] = \frac{1}{\sqrt{\alpha_t}} \big( x_t + (1 - \alpha_t) \nabla \log p_t(x_t) \big). \tag{9}$$

Crucially, the posterior covariance is related to the local curvature of the log-density, *i.e.*, $\nabla^2 \log p_t(x_t)$. Because these quantities are defined by the data distribution and noise schedule rather than the specific model architecture, modern backbones (*e.g.*, U-Net, DiT) trained to approximate the same target score $\nabla \log p_t(x_t)$ exhibit statistically consistent estimation errors and uncertainty structures.

We note that modern diffusion/flow-matching frameworks (e.g., EDM, FLUX) increasingly adopt straightened transport paths and deterministic sampling (*i.e.*, solving ODEs via Euler or Heun methods) for stable generation. Under such deterministic ODE trajectories, inversion and reconstruction are primarily governed by the learned vector field (drift) rather than stochastic noise injection. Consequently, the sensitivity to exact covariance structures is significantly reduced, mitigating residual variance discrepancies across different models. Thus, Assumption B.1 is not merely a mathematically convenient simplification, but a realistic reflection of practical model standardization.

Second, we define a distortion metric to quantify representation quality. In rate-distortion theory, the choice of distortion measure is critical for determining the theoretical limits of compression. Under the Gaussian assumption, mean squared error (MSE) is both standard and analytically tractable. We thus adopt MSE to assess the fidelity of recovered latents.

**Assumption B.2** (Mean Squared Error Distortion). The latent recovery quality is measured by the normalized mean squared error (MSE) between the original latent variable $X_0$ and the adversary's estimated latent $\hat{X}_0$:

$$D(\hat{X}_0, X_0) = \mathbb{E}\left[\frac{1}{d}\|X_0 - \hat{X}_0\|_2^2\right],$$

where the expectation is over the joint distribution of $(X_0, \hat{X}_0)$.

## B.2. Proof of Theorem 4.2

*Proof.* We derive the lower bound under the Gaussian latent approximation and MSE distortion.

**Step 1: Proxy-target Posterior Mismatch.** Let $P = P_{\Theta}(\cdot \mid Y_w), Q = P_{\Theta_{\mathcal{A}}}(\cdot \mid Y_w)$ denote the target and proxy posteriors. By Def. 4.1, the mismatch-induced irreducible information loss is $D_{\mathrm{irr}} = \mathbb{E}_{Y_w}[D_{\mathrm{KL}}(P\|Q)]$. When $D_{\mathrm{irr}} > 0$, the proxy posterior differs from the target posterior. This posterior mismatch captures an irreducible source of reconstruction error that cannot be removed by increasing the information rate. Under our Gaussian rate-distortion surrogate, this mismatch contributes to the additive term $D_{\mathrm{irr}}$ in the distortion lower bound.

**Step 2: Effective Information Rate.** Since the adversary reconstructs the latent through the proxy posterior rather than the target posterior, only part of the nominal information rate $R$ is usable for target-consistent reconstruction. Under the Gaussian latent approximation, we model this loss by the effective rate penalty defined in Def. 4.1,

$$I_{\text{pen}} = \frac{1}{2} \log_2 \left(1 + \frac{D_{\text{irr}}}{\sigma^2}\right).$$

Thus, the adversary operates at an effective rate $R_{\text{eff}} \leq R - I_{\text{pen}}$.

**Step 3: Gaussian Rate-distortion Lower Bound.** For a Gaussian source with variance $\sigma^2$ under MSE distortion, the inverse rate-distortion function is $D(R) = \sigma^2 2^{-2R}$. Since $D(R)$ is monotonically decreasing in $R$, substituting $R_{\text{eff}} \leq R - I_{\text{pen}}$ gives

$$D(R_{\text{eff}}) \geq \sigma^2 \cdot 2^{-2(R - I_{\text{pen}})}.$$

Since $D_{\text{irr}} > 0$, the additive mismatch term prevents the lower bound from vanishing, while $I_{\text{pen}} > 0$ further reduces the effective rate. This establishes the theorem. $\square$

**Relation to Total Variation.** The KL mismatch above also implies distributional separation between the target and proxy posteriors. When $D_{\text{irr}} > 0$, the target and proxy posteriors are distinct on a set of nonzero measure. Pinsker's inequality provides the standard upper control $\|P - Q\|_{\text{TV}} \leq \sqrt{\frac{1}{2} D_{\text{KL}}(P\|Q)}$. This relation is used only to interpret posterior mismatch as a distributional discrepancy. The rate penalty in Thm. 4.2 is the Gaussian surrogate defined in Def. 4.1.

### B.3. Proof of Theorem 4.10

The proof of Theorem 4.10 leverages two complementary geometric perspectives. Lemma 4.6 establishes the existence of a persistent global angular shift on the latent hypersphere, whereas Lemma 4.9 accounts for the local structural inconsistency on the SPD manifold.

*Proof of Lemma 4.6.* We analyze the geometric effect of a proxy-induced perturbation on the hyperspherical representation of the latent. Let $z \equiv z_T^{(w)} \sim \mathcal{N}(0, \mathbf{I}_N)$ and consider a forged recovered latent $\hat{z} = z + \delta$.

We begin by decomposing the perturbation $\delta$ into components parallel and orthogonal to $z$:

$$\delta = \delta_\| + \delta_\perp, \qquad \delta_\| = \alpha z, \quad \delta_\perp \perp z.$$

By construction, the parallel component affects only the radial magnitude, while the orthogonal component governs orientation changes. The angular distortion between $z$ and $\hat{z}$ is given by

$$\cos \theta = \frac{\langle z, z + \delta \rangle}{\|z\|_2 \|z + \delta\|_2}.$$

Using the above decomposition and noting that $\langle z, \delta_\perp \rangle = 0$, we obtain

$$\langle z, z + \delta \rangle = \|z\|_2^2 + \langle z, \delta_\| \rangle,$$

and

$$\|z + \delta\|_2^2 = \|z\|_2^2 + 2\langle z, \delta_\| \rangle + \|\delta_\|\|_2^2 + \|\delta_\perp\|_2^2.$$

Assuming $\|\delta\|_2 \ll \|z\|_2$, a first-order Taylor expansion of the denominator yields

$$\cos \theta \approx 1 - \frac{\|\delta_\perp\|_2^2}{2\|z\|_2^2}.$$

This approximation shows that the induced angular deviation depends solely on the orthogonal component of the perturbation.

Finally, since $z \sim \mathcal{N}(0, \mathbf{I}_N)$, its norm concentrates sharply around $\|z\|_2^2 \approx N$ as $N \to \infty$. Under the proxy-target model mismatch, the perturbation $\delta$ cannot be consistently aligned with $z$ in expectation, implying that the orthogonal energy

$\|\delta_\perp\|_2^2$ remains strictly positive. Consequently, the angular distortion is bounded away from zero, acting as a manifest signature of forgery. Substituting into the above expression yields

$$\mathbb{E}[\text{SAD}(z, \hat{z})] = \mathbb{E}[\theta] = \Omega(1),$$

which completes the proof of Lemma 4.6. □

*Proof of Lemma 4.9.* In this proof, we exploit the *congruence invariance* of the AIRM, which states that for any $P, Q \in \mathcal{P}_N$ and any invertible matrix $G \in \mathbb{R}^{N \times N}$, the distance satisfies $d_{\text{AIRM}}(P, Q) = d_{\text{AIRM}}(GPG^\top, GQG^\top)$.

Let $P := \mathcal{C}(z_T^{(w)}) \in \mathcal{P}_N$ denote the local covariance associated with the reference latent $z_T^{(w)}$. To derive the probability bound, we analyze the geometric deviation in two distinct cases:

**Case 1: Watermarked recovered latents.** For a watermarked recovered latent $\hat{z}_T^{(w)}$ produced by the target model, the relative configuration of tokens within the local neighborhood $\mathcal{B}$ is preserved up to negligible estimation error. Thus, the induced covariance $Q^{(w)} := \mathcal{C}(\hat{z}_T^{(w)})$ remains close to $P$ in the SPD manifold.

Since the regularization $\varepsilon \mathbf{I}$ ensures strict positive definiteness, the AIRM metric is continuous on $\mathcal{P}_N$. Therefore, such structural preservation under an affine (congruent) transformation yields:

$$\text{LGI}(z_T^{(w)}, \hat{z}_T^{(w)}) = d_{\text{AIRM}}(P, Q^{(w)}) \approx 0.$$

**Case 2: Forged recovered latents.** Consider a forged recovered latent $\hat{z}_T^{(f)} = \hat{z}_T^{(w)} + \delta$ that induces a *non-congruent deformation* of the local token neighborhood $\mathcal{B}$. As a result, the induced covariance $Q^{(f)}$ cannot be expressed as $GPG^\top$ for any invertible matrix $G$ sufficiently close to the identity, and thus lies outside the local congruence orbit of $P$ on the SPD manifold.

Mathematically, let $S := P^{-1/2} Q^{(f)} P^{-1/2}$. The structural mismatch implies that $S$ cannot be close to the identity. Specifically, there exists a constant $\eta_0 > 0$ such that, with non-vanishing probability $p_0 > 0$, the spectrum of $S$ satisfies:

$$\lambda_{\max}(S) \geq e^{\eta_0} \quad \text{or} \quad \lambda_{\min}(S) \leq e^{-\eta_0}.$$

Recall that the AIRM geodesic distance is characterized by the log-eigenvalues of $S$. Thus, by the property of the Frobenius norm:

$$\text{LGI}(z_T^{(w)}, \hat{z}_T^{(f)}) = \| \log S \|_F = \sqrt{\sum_i \ln^2 \lambda_i(S)} \geq \max_i |\ln \lambda_i(S)| \geq \eta_0.$$

Finally, combining the probability bound $p_0$ with the spectral gap $\eta_0$, and taking the expectation over the dataset, we obtain:

$$\mathbb{E}\Big[\text{LGI}(z_T^{(w)}, \hat{z}_T^{(f)})\Big] \geq \eta_0 \, p_0 = \Omega(1).$$

This completes the proof of Lemma 4.9. □

*Proof of Theorem 4.10.* The proof relies on a concentration-of-measure argument in high-dimensional spaces, combining the global geometric drift established in Lemma 4.6 and the local structural deformation from Lemma 4.9.

First, we define a rejection region $\mathcal{R} \subset \mathbb{R}_+^2$ in the joint metric space defined by SAD and LGI:

$$\mathcal{R} := \{(s, \ell) \in \mathbb{R}_+^2 \mid s > \tau_s \text{ and } \ell > \tau_\ell\},$$

where $\tau_s$ and $\tau_\ell$ are positive constants.

● *Forged recovered latents*: By Lemma 4.6, the expected angular distortion induced by the orthogonal perturbation component $\|\delta_\perp\|$ is bounded away from zero (*i.e.*, $\mathbb{E}[\text{SAD}] = \mu_s = \Omega(1)$). Similarly, Lemma 4.9 demonstrates that non-congruent structural deformation leads to a non-vanishing local geometric inconsistency (*i.e.*, $\mathbb{E}[\text{LGI}] = \mu_\ell = \Omega(1)$).

However, divergence in expectation alone is insufficient to imply separability. To bridge this gap, we invoke concentration-of-measure results in high-dimensional spaces ($N \gg 1$). Under the assumption that the considered geometric metrics are

Lipschitz continuous *w.r.t* the latent variables, it is a well-established result that for isotropic Gaussian distributions, and their normalized projections onto the hypersphere, such quantities concentrate sharply around their expectations, with tails that decay exponentially in the dimension. Consequently, the probability that the observed distortions fall significantly below their expectations decays exponentially with the dimension $N$.

For thresholds chosen such that $0 < \tau_s < \mu_s$ and $0 < \tau_\ell < \mu_\ell$, we obtain:

$$\Pr(\text{SAD} > \tau_s) \geq 1 - e^{-\Omega(N)}, \quad \text{and} \quad \Pr(\text{LGI} > \tau_\ell) \geq 1 - e^{-\Omega(N)}.$$

Applying a union bound, the forged sample falls within the rejection region $\mathcal{R}$ with high probability:

$$\Pr((\text{SAD}, \text{LGI}) \in \mathcal{R}) \geq 1 - 2e^{-\Omega(N)} = 1 - e^{-\Omega(N)}.$$

• *Watermarked recovered latents*: For a watermarked recovered latent $\hat{z}_T^{(w)}$ generated by the target model, the recovery error is minimal and structurally coherent. As $N \to \infty$, both SAD and LGI converge to zero in probability. Therefore, the probability that a watermarked sample falls within the rejection region $\mathcal{R}$ vanishes asymptotically.

Combining these results, we find that as $N \to \infty$, the overlap probability between the distributions of watermarked and forged latents in the joint geometric space vanishes. Thus, there exists a decision boundary that separates the two classes with probability approaching one. □

## C. Additional Analysis

### C.1. Intrinsic and External Errors in Diffusion Models

**Internal Accumulated Error.** Internal accumulated error $\varepsilon_{\text{int}}$ arises from systematic mismatches between the forward generation and backward inversion processes. In practice, forward diffusion is conditioned on prompts and amplified by classifier-free guidance, while inversion is typically performed under unconditional settings with approximate solvers. Although each stepwise discrepancy is small, such biases accumulate over the diffusion horizon, resulting in a non-negligible estimation error, *internal latent drift*:

$$\varepsilon_{\text{int}} := dist\left(\mathcal{I}_\Theta(z_0^{(w)}), z_T^{(w)}\right),$$

where $dist(\cdot, \cdot)$ denotes a distance metric (*e.g.*, $L_2$ norm) in the latent space, measuring the divergence of the reconstructed latent $\mathcal{I}_\Theta(z_0^{(w)})$ from the ground-truth $z_T^{(w)}$. This intrinsic error is unavoidable and establishes a baseline distortion level (lower bound) for evaluating watermark robustness within the latent domain.

**External Inevitable Perturbations.** Beyond internal drift, we consider two primary external sources of degradation: (*i*) common image-domain distortions $\varepsilon_{\text{img}}$, and (*ii*) black-box forgery-induced model mismatch $\varepsilon_{\text{mis}}$. The latter occurs when a watermarked image is manipulated by an unknown proxy model $\Theta_\mathcal{A}$. We define this mismatch error $\varepsilon_{\text{mis}}$ as the latent deviation arising from the internal processing of $\Theta_\mathcal{A}$:

$$\varepsilon_{\text{mis}} := \left(\mathcal{I}_{\Theta_\mathcal{A}} \circ \mathcal{G}_{\Theta_\mathcal{A}}\right)(z_T^{(w)}) - z_T^{(w)},$$

which characterizes the effective latent deviation induced by $\Theta_\mathcal{A}$. The magnitude of $\varepsilon_{\text{mis}}$ is correlated with the architectural discrepancy between $\Theta_\mathcal{A}$ and $\Theta$: as this discrepancy increases, the induced inversion–sampling mapping departs further from identity, resulting in a larger latent mismatch. Therefore, $\varepsilon_{\text{mis}}$ constitutes an intrinsic, architecture-induced deviation that is unavoidable without access to the exact target parameters $\Theta$.

### C.2. Comparison of Forgery Paradigms and Threat Models

Our study focuses on black-box forgery scenarios, in which an adversary exploits proxy models to regenerate watermarked content. We distinguish this setting from alternative forgery paradigms, such as statistical forgery (Yang et al., 2024a) or watermark copying (Dong et al., 2025), which are based on different assumptions and threat models. Specifically, the statistical averaging method (Yang et al., 2024a) requires access to a collection of watermarked samples in order to estimate the underlying watermark distribution. Such requirements are often impractical in realistic settings, where an adversary typically has limited access to watermarked data. Moreover, existing evaluations primarily focus on frequency-based semantic watermarks (*e.g.*, TR), and their effectiveness against bitstream-level schemes (*e.g.*, GS) remains unclear. In

contrast, WMCopier (Dong et al., 2025) transplants invisible signatures onto arbitrary images through unconditional diffusion and iterative optimization. However, its evaluation mainly targets post-processing watermarks (Cox et al., 2008), rather than the semantic watermarking schemes (*e.g.*, TR and GS) considered in our work.

To the best of our knowledge, black-box forgery (Müller et al., 2025) remains the only approach demonstrated to produce forged outputs against semantic watermarking schemes successfully.

### C.3. Discussion on Adaptive Attacks

The adversary may be aware of the deployment of anti-forgery detection and could develop adaptive attacks to circumvent it. We consider one possible such adaptive attack, *Reforge*, described below.

**Reforge.** *Reforge* adopts a two-stage adaptive strategy: cleansing and re-imprinting. The adversary first removes the original watermark through purification and then imprints an estimated watermark pattern using the proxy model. This two-stage process aims to deceive the watermark detector while preserving the semantic content of the original image.

In practice, such operations inevitably lead to perceptual degradation (*e.g.*, reduced PSNR and SSIM), as removal attacks typically require additive noise to drive the latent representation away from the reference watermarked latent $z_T$. The subsequent re-imprinting further shifts the latent toward the watermarked latent, but this manipulation compounds the structural damage, severely compromising image fidelity with noticeable visual artifacts.

We exclude *Reforge* from our comparison due to its prohibitive computational overhead and poor image fidelity. Furthermore, this two-stage adaptive strategy, which relies on a proxy model, significantly distorts the underlying geometric structure of the original latent manifold. Thus, our metric remains effective against such adaptive attempts.

### C.4. Our Detection Pipeline

As shown in Fig 8, the detection metrics are computed between the reference watermarked latent $z_T$ and the recovered latent $\hat{z}_T$ before watermark verification. Importantly, our detection framework is agnostic to the specific watermarking scheme and can be generalized to any latent-based (*i.e.*, semantic) watermarking that utilizes deterministic reverse diffusion trajectories.

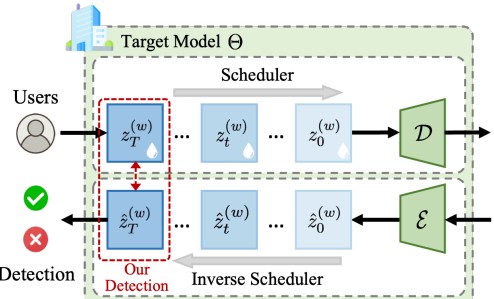

*Figure 8.* Illustration of the proposed detection framework.

### C.5. Connection between Removal Guarantees and Distortion Bounds

Recent work (Zhao et al., 2024) theoretically analyzes the removability of invisible watermarks under stochastic regeneration perturbations in diffusion models. Although that work focuses on post-hoc watermarking, whereas our work studies in-generation watermark schemes, both are related through the distortion introduced during watermark manipulation.

Specifically, (Zhao et al., 2024) shows that successful watermark removal requires sufficient stochastic perturbation, which progressively degrades reconstruction fidelity. In contrast, our analysis reveals that black-box forgery remains subject to $D_{\mathrm{irr}}$ caused by proxy-target mismatch and inversion-generation asymmetry, even without explicit stochastic perturbation. We further find that this distortion manifests geometrically as directional drift on the hypersphere and localized deformation on the SPD manifold. These observations indicate a fundamental trade-off among watermark suppression, forgery fidelity, and geometric consistency in generative models.

# D. Experimental Details

In this section, we provide a comprehensive overview of experiments. We elaborate on diffusion model architectures, the datasets, and the computational runtime. Besides, we present additional experiments to provide further insights.

## D.1. Target and Proxy Models

We evaluate our method across diverse diffusion model architectures, with target models, including SD2.1, SDXL, PixArt-$\Sigma$, SD3, and FLUX.1. Tab. 6 summarizes the sampling configurations and key hyperparameters for all models used in our experiments. Notably, FLUX.1 and SD3, which are based on rectified-flow matching, employ fewer inference steps (*i.e.*, 20 for FLUX.1) and a lower guidance scale (*i.e.*, 7.0 for SD3).

*Table 6.* Overview of model settings used in the experiments. All images are generated at a $512 \times 512$ resolution. *L. Ch.* denotes the number of latent channels; *Scheduler* indicates the algorithm used for generation and inversion; *Steps* refers to the number of inference steps; *G. Scale* represents the guidance scale during generation.

| Model | Hugging Face ID | Type | L. Ch. | Scheduler | Steps | G. Scale |
|-------|-----------------|------|--------|-----------|-------|----------|
| SD2.1 | stabilityai/stable-diffusion-2-1-base | UNet | 4 | DDIM | 50 | 7.5 |
| SDXL | stabilityai/stable-diffusion-xl-base-1.0 | UNet | 4 | DDIM | 50 | 7.5 |
| Pixel-$\Sigma$ | PixArt-alpha/PixArt-Sigma-XL-2-512-MS | DiT | 4 | DPM | 20 | 4.5 |
| FLUX.1 | black-forest-labs/FLUX.1-dev | DiT | 16 | FlowMatchEuler | 20 | 3.5 |
| SD3 | stabilityai/stable-diffusion-3-medium | DiT | 16 | FlowMatchEuler | 28 | 7.0 |

## D.2. Prompting and Cover Image Datasets

We employ the Stable-Diffusion-Prompt (SDP)[2] dataset to generate the watermarked images from all target models. For guidance-based forgery attacks, we draw prompts from the same dataset but select a disjoint set to re-guide image generation, ensuring that the forged images are visually distinct from the corresponding watermarked ones. Importantly, we intentionally maintain a consistent prompt distribution between the service provider and the adversary, as this setting provides the adversary with a stronger and more realistic forgery capability. Accordingly, all experiments are conducted within the SDP dataset, rather than using an external prompt source (*i.e.*, the Inappropriate Image Prompts (I2P)[3] dataset adopted in (Müller et al., 2025)). For optimization-based attacks, we adopt the MS-COCO-2017 Dataset (Lin et al., 2014) as cover images, consistent with (Müller et al., 2025).

## D.3. Runtime of Attack Algorithms

The experiments described in Sec. 5 are conducted on a single A6000 GPU, and all methods are evaluated under identical system conditions. To evaluate computational overhead, we report the approximate per-sample execution time for each attack, conducted on a single GPU in a single-batch configuration:

• The optimization-based forgery attacks (Sec. A.2) require approximately between 15 and 20 minutes to complete 100 steps. The most time-consuming part of these algorithms is the gradient-based optimization done by the adversary's model (SD2.1 by default). In contrast, verification by the target model, conducted after 20, 50, or 100 optimization steps, is comparatively fast and incurs negligible overhead.

• The guidance-based attack (Sec. A.2) requires between 30 seconds for smaller models (*i.e.*, SD2.1, SDXL, PixArt-$\Sigma$), and 4 minutes for larger models such as FLUX.1 and SD3. This time accounts for all stages of the attack, including generating a watermarked image with the target model, inverting and regenerating it using the adversary's model, and verifying the presence of the watermark with the target model.

## D.4. Distortion Methods and Parameters

To evaluate the robustness of our detection framework, we apply 14 commonly used post-processing distortions to the watermarked images before the inversion process. These distortions are designed to emulate realistic image modifications that may interfere with watermark extraction. Visual examples of each attack type are provided in Fig. 9.

---

[2]Stable-Diffusion-Prompts
[3]Inappropriate Image Prompts (I2P)

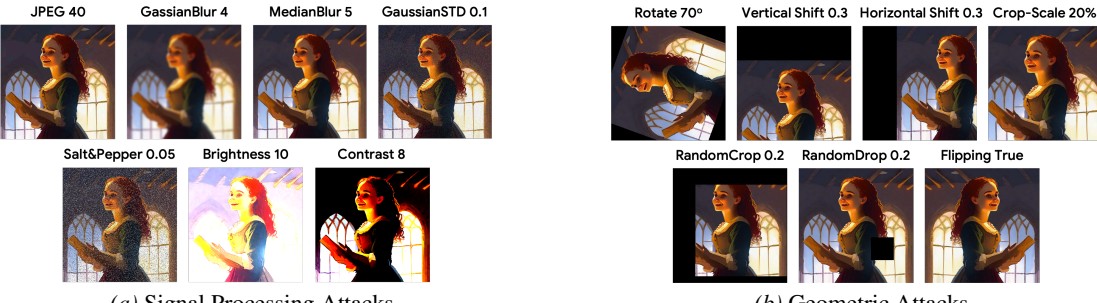

*(a)* Signal Processing Attacks          *(b)* Geometric Attacks

*Figure 9.* Visual examples of the post-processing distortions used in our robustness evaluation. (*a*) illustrates signal processing distortions such as JPEG compression levels and noise intensities, while (*b*) shows geometric and erasure-based attacks.

- *Signal Processing Attacks*: We consider a broad spectrum of signal distortions with varying intensities to simulate realistic transmission and editing artifacts. These include brightness and contrast adjustments, JPEG compression, Gaussian and Median blurring, and additive noise (*i.e.*, both Gaussian and salt-and-pepper types). We examine these attacks across the parameter ranges specified in Fig. 7 of the main text.

- *Geometric Attacks*: We further consider a range of geometric transformations that alter the spatial structure of images while largely preserving semantic content. These include scaling, cropping, horizontal/vertical translation, flipping, rotation, and random region dropping (occlusion). Parameter ranges for each geometric attack are specified in Fig. 7 of the main text.

### D.5. Empirical Validation of Theoretical Assumptions

We empirically examine Assumption B.1 by analyzing the inversion residuals $(\hat{z}_T^{(f)} - z_T^{(w)})$ under the reprompting forgery attack with the TR scheme. To assess the Gaussian latent approximation, we apply a random-projection normality test to the residuals. As shown in Tab. 7, across different target models with SD2.1 as the proxy, the average skewness and kurtosis remain close to 0 and 3, respectively. This indicates that the residual statistics are broadly consistent with a Gaussian approximation.

*Table 7.* Normality test of inversion residuals (Proxy: SD2.1).

| Target | Skewness | Kurtosis |
|---|---|---|
| SD2.1 | 0.0092 | 3.1694 |
| SDXL | 0.0015 | 3.0272 |
| PixArt-$\Sigma$ | -0.0083 | 3.0050 |
| FLUX | -0.0040 | 3.0612 |
| SD3 | -0.0106 | 3.0539 |

We further examine the shared-variance premise by comparing latent posterior variances across model architectures. The measured variance ratios are $1.2265$ for SDXL/SD2.1 and $0.9508$ for PixArt-$\Sigma$/SD2.1, both close to the ideal ratio of $1.0$. We also observe nontrivial per-dimension correlations of $0.4421$ and $0.3496$, respectively. These results suggest that, although the architectures differ, their latent uncertainty structures remain sufficiently aligned to support the tractable shared-variance approximation used in our analysis.

### D.6. Additional Experiments

**Distributional Analysis under Guidance-based Attacks.** We analyze the distributions of unwatermarked, watermarked, and forged samples under guidance-based attacks to understand the geometric separability induced by proxy-based generation.

- SDXL (Target) $\rightarrow$ SD2.1 (Proxy): As shown in Fig. 10, the distributions of unwatermarked (blue) and watermarked samples (green) are nearly identical, indicating that the semantic watermark preserves the original latent distribution, *i.e.*, a Gaussian distribution, and maintains high perceptual fidelity. However, forged images generated by the proxy model inevitably introduce structural perturbations in the latent space, enabling forged samples to be clearly distinguished from watermarked ones, even under a similar target–proxy model setting.

- FLUX (Target) $\rightarrow$ SD3 (Proxy): We further report the distribution between watermarked and forged samples under the

flow matching model. As shown in Fig. 11, the two distributions are clearly separable under the local SPD distance. When FLUX is used as the target model, the distribution is broader, whereas the SD3-based distribution is more concentrated. A similar separation is observed in Fig. 12 using cosine similarity. Notably, FLUX-generated samples exhibit a wider cosine similarity range (*i.e.*, $0.4$ to $0.8$), forming a longer tail compared to the range (*i.e.*, $0.6$ to $0.8$) reported in Tab. 5.

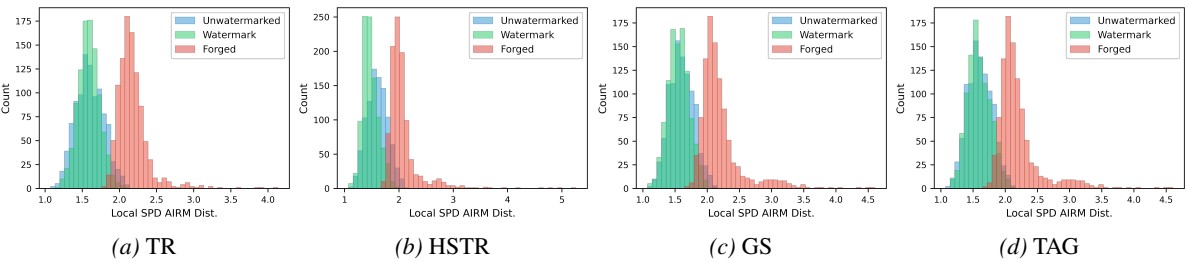

| *(a)* TR | *(b)* HSTR | *(c)* GS | *(d)* TAG |

*Figure 10.* Distributions of local SPD (AIRM) distances for unwatermarked, watermarked, and forged samples. The results are shown for SDXL as the target model and SD2.1 as the proxy.

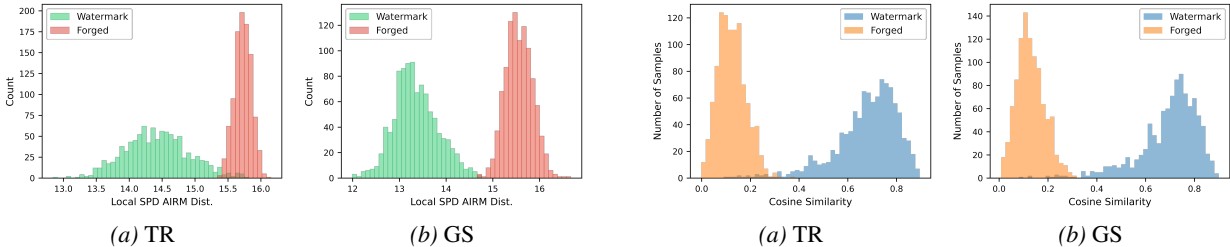

| *(a)* TR | *(b)* GS | | *(a)* TR | *(b)* GS |

*Figure 11.* Distributions of local SPD distances for watermarked and forged samples, under the FLUX (target) and SD3 (proxy).

*Figure 12.* Distributions of cosine similarity for watermarked and forged samples. The target-proxy setup is consistent with Fig. 11.

**Hyperparameter Sensitivity of L-SPD.** To evaluate the robustness of L-SPD to hyperparameter choices, we conduct a sensitivity analysis under the SDXL (target) and SD2.1 (proxy) setting. As shown in Table 8, L-SPD exhibits high stability across different grid sizes ($G \in \{4, 8, 16\}$), top-$k$ selections ($k \in \{5, 8, 16\}$), and aggregation methods (Mean, Top-$k$ Mean, and Std). The AUC consistently remains above $0.932$ and reaches up to $1.000$, indicating that the proposed metric captures intrinsic geometric deformation rather than relying on specific hyperparameter configurations.

*Table 8.* Hyperparameter sensitivity analysis of L-SPD.

| Group | Parameter Variation | TR (AUC) | GS (AUC) |
| --- | --- | --- | --- |
| Default | $G = 16, k = 5$, Top-$k$ Mean | 0.997 | 0.995 |
| Grid Size | $G \in \{4, 8, 16\}$ | 0.932/0.983/0.997 | 0.963/0.995/0.995 |
| Patch Top-$k$ | $k \in \{5, 8, 16\}$ | 0.997/0.999/1.000 | 0.995/0.998/0.999 |
| Aggr. | $\{$Mean, Top-$k$ Mean, Std$\}$ | 1.000/0.997/0.997 | 1.000/0.995/0.994 |

**Detection Performance via MSE Metric.** As discussed in Sec. 5.4, the MSE metric can be regarded as a quantitative indicator of feature-level distortion, but remains insufficient for robust detection across a wide range of image distortions. As shown in Fig. 13, the latent MSE distributions exhibit substantial overlap under standard distortions (*e.g.*, Cropping, Gaussian Blurring). Moreover, increasing discrepancies between proxy and target models further shift the MSE distributions toward higher values (red arrow), complicating the selection of reliable detection thresholds.

**Robustness Analysis on PRCW.** As discussed in Sec. 6.1, the PRCW scheme exhibits inherent robustness against guidance-based forgeries in cross-model settings, stemming from its undetectable design, which prevents the adversary from accurately estimating the watermarked latent through a proxy model. However, this robustness degrades substantially when the adversary employs a proxy model identical to the target (*i.e.*, to $0.974$ for SD2.1 and $0.527$ for SD3); in such cases, PRCW becomes highly susceptible to successful forgery, as reported in Tab. 9. In contrast, our proposed method remains effective (*i.e.*, G-Cos $= 0.977$ for SD2.1 and $0.931$ for SD3) even under this stronger threat model, as shown in Tab. 10. This implies that our latent-drift analysis successfully captures the fundamental discrepancy between forged and watermarked

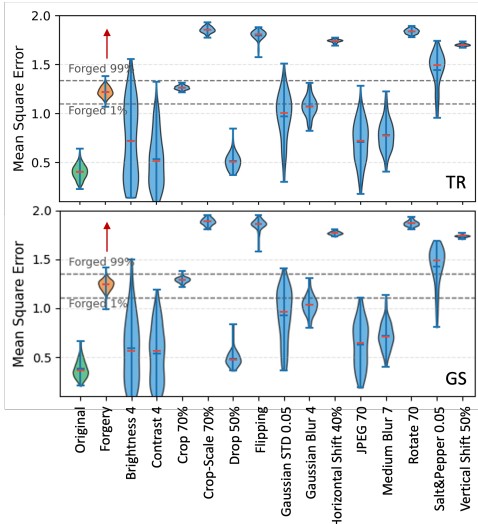

*Figure 13.* MSE distributions under 14 types of image distortions for TR and GS, evaluated on SDXL as the target model.

samples, enabling robust detection across a broader range of attack scenarios. We omit the analysis of optimization-based attacks here, as they exhibit marginal attack efficacy compared to guidance-based methods.

*Table 9.* PRCW TPR under guidance-based attacks (TPR@$X$-FPR). The metric is consistent with those described in Tab. 1.

| | SD2.1 (Proxy) | SD3 (Proxy) |
|---|---|---|
| Target | Det. | Det. |
| SD2.1 | 0.974 | 0.117 |
| SDXL | 0.003 | 0.012 |
| PixArt-$\Sigma$ | 0.024 | 0.027 |
| FLUX | 0.000 | 0.008 |
| SD3 | 0.000 | 0.952 |

*Table 10.* Our detection performance (AUC) under optimization-based attacks. "G-Cos" and "L-SPD" are used here as in the guidance-based attack scenarios (see Tab. 2).

| Proxy | Target | G-Cos | L-SPD |
|---|---|---|---|
| SD2.1 | SD2.1 | 0.977 | 0.951 |
| SD3 | SD3 | 0.931 | 0.700 |

# E. Example of Forgery Images

In this section, we provide additional example images generated by guidance- and optimization-based forgery methods. All images are generated at a resolution of $512 \times 512$, and the adversary employs SD2.1 as the proxy model.

### E.1. Guidance-based Methods

Fig. 14 presents a visual comparison between watermarked images produced by different target models and forged images generated via the proxy model using GS and TR schemes.

### E.2. Optimization-based Methods

Fig. 15 illustrates the progression of the imprinting forgery attack as the number of optimization steps increases. As discussed in (Müller et al., 2025), successful forgeries typically require at least 20 optimization steps during which the accumulation of adversarial noise severely degrades image fidelity (*e.g.*, with PSNR values dropping below 24 dB).

Guidance-based Attack

| | SD2.1 | | SDXL | | PixArt-Σ | | FLUX.1 | | SD3 | |
|---|---|---|---|---|---|---|---|---|---|---|
| | TR | GS | TR | GS | TR | GS | TR | GS | TR | GS |

" young, curly haired, redhead Natalie Portman as a optimistic!, cheerful, giddy medieval innkeeper in a dark medieval inn. dark shadows, colorful, candle light, law contrasts, fantasy concept art by Jakub Rozalski, Jan Matejko, and J.Dickenson "

" portrait of a young ruggedly handsome but joyful pirate, male, masculine, upper body, red hair, long hair, d & d, fantasy, smirk, intricate, elegant, highly detailed, digital painting, artstation, concept art, matte, sharp focus, illustration, art by artgerm and greg rutkowski and alphonse mucha "

*Figure 14.* Examples of guidance-based attacks on different target models and watermarking schemes. The top row shows watermarked images, generated using prompts from the SDP dataset with the target model indicated at the top of each panel. The subsequent rows present the corresponding forged outputs, where the adversary employs SD2.1 and SD3 as proxy models to perform forgery.

Optimization-based Attack - Progression of Perturbations

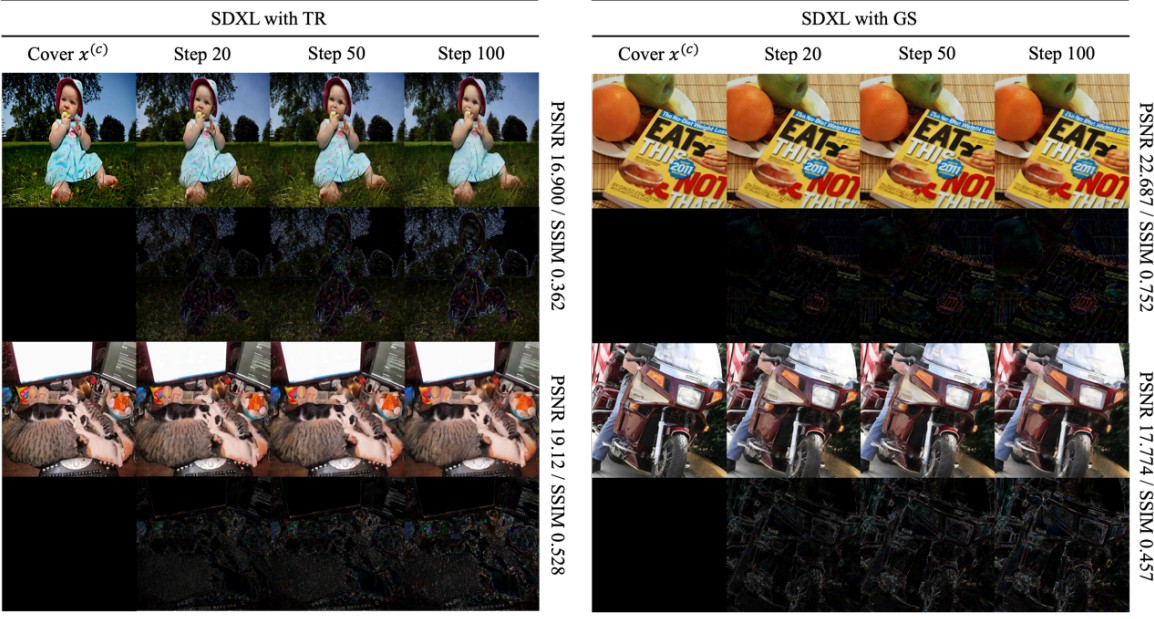

*Figure 15.* Progression of optimization-based forgery attack as the number of optimization steps increases. The target model is SDXL, and the forged watermark is TR and GS. Results are conducted on SDXL with TR and GS watermarks and four cover images $x^{(c)}$. For each cover image, the top row shows the initial cover and the corresponding forged images at different optimization steps, while the bottom row illustrates the absolute pixel-wise differences from the initial cover.

