# OpenReview forum: "Rethinking Forgery Attacks on Semantic Watermarks in Black-Box Settings: A Geometric Distortion Perspective"
_ICML.cc/2026/Conference — ICML 2026 regular_

### Official Review · Reviewer_Y41H · 2026-03-03

**Soundness:** 3
**Presentation:** 3
**Significance:** 2
**Originality:** 3
**Overall Recommendation:** 5
**Confidence:** 4

**Summary:**

This paper employs semantic watermarks to address the security issues under black-box attacks, and models this process as a rate-distortion trade-off problem in the latent space. Based on this, the author proposed a pre-validation method for forged samples that is independent of the watermarking scheme. The experiment proved its effectiveness.

**Compliance With Llm Reviewing Policy:**

Affirmed.

**Final Justification:**

I thank the authors for their detailed responses and the effort put into the revision. After careful consideration, I have decided to upgrade my  score.

**Key Questions For Authors:**

1.In a real scenario, it is entirely possible for an attacker to use a proxy model with the same family and architecture as the target model, or even one with high similarity obtained through distillation or copying. The performance of the proposed method deteriorates obviously in such proxy scenarios. How do the authors view the impact of this problem on the actual security value of the method? Are there further experiments to illustrate the bounds of applicability of this method under stronger attacker Settings?
2.This paper explains why MSE alone is not enough to effectively detect forged samples, but is it enough to show that G-Cos and L-SPD are the most reasonable choices? Have the authors compared simpler but natural alternatives, such as latent anomaly detection, local cosine heatmap aggregation, or other standard anomaly detection/manifold outlier baselines?
3.This method relies on multiple implementation details and hyperparameter choices. Can the authors add: How much do the results fluctuate with different patch/grid partitions, different top-k Settings, and different aggregation methods? Are current methods sensitive to hyperparameters?

**Limitations:**

(1) The proposed method is essentially a pre-verification detection module, which can identify whether the sample has suspicious signs of regeneration/forgery, but it does not further discuss how to deal with the case when the pre-verification results are inconsistent with the formal watermark verification results. It also lacks subsequent disposal mechanism design, so its end-to-end deployment value is still not clear.

(2) The effectiveness of the proposed method is based on the geometric distortion caused by the proxy-target mismatch, so the detection ability may be significantly reduced when the attacker uses a proxy model that is highly similar to the target model. Since this is one of the most serious attack scenarios in reality, the discussion of this security boundary is still insufficient.

**Strengths And Weaknesses:**

**Strengths:**
1.The motivation and approach of this article are innovative. Instead of treating black-box forgery attacks as a form of disturbance, it explains some inherent flaws of such attacks and designs solutions based on those flaws.
2.This method is a plug-and-play plugin that can be used in various scenarios, making it practical.
3.The paper is well-structured with clear narration and a complete overall framework. There are no contradictions between the different parts, and the experimental scope is also quite extensive.

**Weakness:**
1.In the real world, it is very possible to use proxy models with high similarity from the same family, same architecture, or even distillation/replication. As a content security paper, poor performance in this regard is a shortcoming.
2.The paper explains why pure MSE is not enough for detection, but is it not sufficient to show that G-Cos and L-SPD are the most reasonable choices if simpler methods such as latent anomaly detection and local cosine heatmap aggregation have been compared?

---

> ### Author Rebuttal · Authors · 2026-03-29
>
> Dear Reviewer Y41H,
>
> Thank you for your thoughtful and detailed review. We hope our responses resolve your concerns.
>
> > Weakness #1 & Question #1: Stronger attacker setting (same proxy–target models).
>
> We thank the reviewer for the insightful comment. We agree that the identical target-proxy setting is a critical security boundary, where geometric distortion becomes minimal, and detection is more challenging.
>
> To evaluate this scenario, we consider **identical model** pairs in our experiments. As shown in Table 2 of the paper, even when the attacker uses the same architecture as the target (e.g., SD2.1 → SD2.1), the method maintains **clear detection capability**. G-Cos achieves AUC **> 0.955**, and L-SPD exceeds 0.928. For the SD3 family, the AUC remains **above 0.940 in most cases**.
>
> Table 4 of the paper further shows that this performance is **stable under optimization-based attacks**. These results suggest that even under this strong attacker setting, forged samples exhibit measurable geometric discrepancies in the recovered latent. Although weaker than in the mismatched case, they remain detectable, **increasing the attack cost and uncertainty**.
>
> In addition, our current L-SPD uses fixed, task-agnostic hyperparameters (e.g., a 16×16 grid with top-5 mean). Under the TR scheme with reprompting attack in the same-model setting (SD2.1 → SD2.1), tuning these parameters (e.g., top-16 mean) improves AUC from **0.929** to **0.943**. This indicates that the method is effective under default settings and can be further improved with tuning. We will add a comprehensive ablation in the revision.
>
> > Weakness #2 & Question #2: Comparison with Alternative Metrics and Baselines.
>
> We thank the reviewer for this question. We clarify that our framework is metric-agnostic, as any measure capturing the identified geometric deviations can be used, provided it preserves robustness under benign image distortions. G-Cos and L-SPD are not ad hoc choices but empirical realizations of our theoretical results: G-Cos corresponds to SAD (Lemma 4.5), and L-SPD captures LGI (Lemma 4.8).
>
> We further compare with a local cosine baseline (grid size 16, mean). It achieves AUC = 1.00 under model mismatch (SDXL → SD2.1) and 0.954 (TR) / 0.976 (GS) in the same-model setting (SD2.1 → SD2.1). This shows that while simpler alternatives are also effective within our framework, capturing geometric discrepancies is the key to detection. We will include these results in the revision.
>
> > Question #3: Hyperparameter Robustness.
>
> We thank the reviewer for this question. To evaluate robustness, we conduct a sensitivity analysis on the SDXL (target) and SD2.1 (proxy) setting across both TR and GS schemes. As shown in Table 1 below, L-SPD exhibits stability under varying grid sizes and patch selection, indicating **low sensitivity** to hyperparameters and reliable performance under default settings.
>
> Table 1. Hyperparameter sensitivity analysis of L-SPD.
>
> | Group | Parameter Variation | TR (AUC) | GS (AUC) |
> | :--- | --- | :---: | :---: |
> | Default | G=16, k=5, Top-k Mean | 0.997 | 0.995 |
> | Grid Size | G $\in$ \{4, 8, 16\} | 0.932/0.983/0.997 | 0.963/0.995/0.995 |
> | Patch Top-k | k $\in$\{5, 8, 16\} | 0.997/0.999/1.000 | 0.995/0.998/0.999 |
> | Aggr. | \{Mean, Top-k Mean, Std\} | 1.000/0.997/0.997 | 1.000/0.995/0.994 |
>
> > Limitation #1: Decision Conflicts & System Deployment
>
> We thank the reviewer for the valuable comment. If our method detects significant geometric distortion while the watermark verification succeeds, this does not indicate a conflict; rather, it reflects a forgery scenario where the watermark is preserved, but the content is altered.
>
> In practice, the pre-verification stage serves as an early warning mechanism: samples with detected distortion can be flagged or filtered before watermark verification, reducing the risk of accepting forgeries. Our method does not require modifying the underlying watermarking scheme and can be seamlessly integrated into existing pipelines (Sec. 6.3), enabling practical deployment with improved robustness. We will include a brief discussion of such cases in the revision.
>
> > Limitation #2: Discussion on the Security Boundary under Stronger Attackers.
>
> We thank the reviewer for this comment. We agree that the setting where the proxy and target models are identical represents a critical security boundary. To address this, we will include a dedicated discussion in the revision (Sec. 6.4: Security Boundaries under Stronger Attackers).
>
> In this section, we connect the empirical estimation of the minimal $D_\mathrm{irr}$ (Sec. 5.4) with the observed detection performance (Tables 2 and 4) under this extreme scenario. We further analyze this case from a security perspective, showing that our approach provides a complementary defense when watermarking schemes that are undetectable (e.g., PRCW) fail under matched model conditions (Sec. 6.1).
>
> We would be glad to further clarify or expand this discussion if helpful.

---

> > ### Author Rebuttal · Reviewer_Y41H · 2026-04-02
> >
> > I thank the authors for their detailed responses and the effort put into the revision. After careful consideration, I have decided to upgrade my  score.

---

> > > ### Author Response · Authors · 2026-04-04
> > >
> > > Dear Reviewer Y41H,
> > >
> > > Thank you for your careful review and insightful follow-up comments. We have made efforts to thoroughly improve our work accordingly and provide responses for each concern here.
> > >
> > > > Reflection of Rebuttal Points in the Current Manuscript
> > >
> > > We thank the reviewer for this comment. We understand your concern that these points must be formally integrated into the manuscript. Below are the exact text additions and their corresponding section numbers that have been updated in our draft:
> > >
> > > >> **Appendix D.5: Hyperparameter Sensitivity of L-SPD.**
> > > >> To evaluate the robustness of our geometric metrics against hyperparameter selections, we conduct a sensitivity analysis on the SDXL (target) and SD2.1 (proxy) setting. As shown in **Table 9**, L-SPD exhibits remarkable stability across varying grid sizes (G∈{4,8,16}), patch selection (k∈{5,8,16}), and aggregation methods (Mean, Top-k Mean, Std). The AUC remains consistently above 0.932 and peaks at 1.000, indicating that the method relies on the intrinsic geometric deformation rather than ad hoc hyperparameter tuning.
> > >
> > > >> **Table 9: Hyperparameter sensitivity analysis of L-SPD.**
> > > >> |Group|Parameter Variation|TR (AUC)|GS (AUC)|
> > > >> |:---|:---|:---|:---|
> > > >> |Default|$G=16,k=5$,Top-$k$ Mean|0.997|0.995|
> > > >> |Grid Size|$G \in\{4,8,16\}$|0.932/0.983/0.997|0.963/0.995/0.995|
> > > >> |Patch Top-$k$|$k\in\{5,8,16\}$|0.997/0.999/1.000|0.995/0.998/0.999|
> > > >> |Aggr.|{Mean,Top-$k$ Mean,Std}|1.000/0.997/0.997|1.000/0.995/0.994|
> > >
> > > We hope these explicit additions resolve your concerns and will ensure that the key points from our previous discussion are clearly reflected in the revised manuscript.
> > >
> > > > Weakness #2 & Question #2: Motivation and Validation of Geometric Metrics
> > >
> > > We totally agree that the motivation and validation of our metrics should be established within the paper. We have added a dedicated paragraph in Sec. 5.2 that explicitly links the proposed geometric metrics to our theoretical lemmas and compares them against simpler baselines. Below is the exact text addition that has been updated in our draft:
> > >
> > > >> **Sec 5.2. Comparison with Alternative Metrics.**
> > > >> While our detection framework is metric-agnostic, G-Cos and L-SPD are *explicitly chosen as empirical realizations* of the theoretical geometric deviations defined in Lemma 4.5 (SAD) and Lemma 4.8 (LGI). We further compare with a local cosine baseline (grid size 16, mean). It achieves an AUC of $1.000$ under cross-model scenarios (e.g., SDXL → SD2.1) and exhibits superior performance in the identical-model setting (e.g., achieving an AUC of $0.954$ for TR and $0.976$ for GS). This indicates that while simpler alternatives are also effective within our framework, capturing geometric discrepancies is the key to detection.
> > >
> > > We hope these additions address the reviewer’s concerns and clarify the theoretical motivation and empirical validation of our geometric metrics.
> > >
> > > > Limitation #2: Discussion on the Security Boundary under Stronger Attackers.
> > >
> > > We have expanded the manuscript to formally characterize this boundary through the underlying mechanism of diffusion inversion, explaining why reconstruction errors arise even under matched proxy–target models. Below is the exact text addition integrated into our revision:
> > >
> > > >> **Sec 6.4. Practical Security Boundaries under Stronger Attackers.**
> > > >> The identical target-proxy setting (e.g., SD2.1 → SD2.1) represents a critical security boundary where proxy-induced architectural mismatch ($\epsilon_{mis}$) is minimized. However, the practical security boundary is strictly lower-bounded by an unavoidable internal accumulated error ($\epsilon_{int}$), as detailed in Appendix C.1. Furthermore, in a black-box setting, these stepwise discrepancies accumulate irreversibly over the diffusion process, ensuring $\epsilon_{int} > 0$ even under perfect architectural alignment. Thus, this irreducible drift precludes exact trajectory reversal and induces structured geometric distortions on the latent manifold, which are captured by our G-Cos and L-SPD metrics. In addition, we note that the pre-verification stage acts as an early warning mechanism to resolve potential decision conflicts. Samples with anomalous geometric drift are flagged as forgeries, preventing them from being accepted even if watermark verification succeeds.
> > >
> > > Once again, we sincerely appreciate the reviewer’s thorough reading of the paper and the time invested in evaluating both the original submission and our revision. Your feedback has helped us significantly improve the clarity and positioning of the work. If there is anything that is still unclear or any additional feedback you would like us to address, we would be very happy to provide further clarification.

---

### Official Review · Reviewer_CudM · 2026-03-10

**Soundness:** 3
**Presentation:** 3
**Significance:** 3
**Originality:** 3
**Overall Recommendation:** 6
**Confidence:** 3

**Summary:**

This paper analyses black box forgery attacks on semantic watermarks in latent diffusion models through a rate distortion framework, identifying an irreducible distortion floor from proxy/target model mismatches. The distortion is characterised as structured geometric deviations (global drift on the hypersphere, local deformation on the SPD manifold), and a scheme agnostic detection method (G Cos + L SPD) is proposed.

**Compliance With Llm Reviewing Policy:**

Affirmed.

**Final Justification:**

Concerns fully resolved (see rebuttal acknowledgement).

**Key Questions For Authors:**

1. Can D_irr be estimated or bounded for specific model pairs (e.g., SD2.1 vs SDXL)? This would make the theoretical contribution more concrete and allow readers to assess how the distortion floor varies across architectures.
2. Has Assumption B.1 been empirically validated? Could the gap between Gaussian and true posteriors be quantified for the studied architectures?
3. (Constructive, does not affect the current assessment) This paper characterises a lower bound on forgery distortion: even with faithful inversion, model mismatch causes irreducible error. Existing work (Zhao et al., 2023, "Invisible Image Watermarks Are Provably Removable Using Generative AI") characterises the opposite end: with sufficient randomness during regeneration, watermarks can be destroyed entirely. These appear to be complementary bounds on a shared distortion continuum. Could connections between the two theoretical perspectives be established?

**Limitations:**

yes

**Strengths And Weaknesses:**

### Strengths

1. The rate distortion framework (Theorem 4.2) provides a principled analysis of the fundamental constraints on black box forgery. The decomposition into a rate independent distortion floor and a rate dependent term with reduced effective rate is clean and well motivated.
2. The geometric characterisation translates the abstract theoretical bound into concrete, measurable signatures (SAD on the hypersphere, LGI on the SPD manifold). Theorem 4.9 on probabilistic geometric separability ties the theory to practical detection.
3. The experiments are comprehensive: 5 target models, 3 proxy models, 6 watermarking schemes, 2 attack types, and robustness analysis against 14 distortion types. Detection AUC exceeds 0.99 in most scenarios.

### Weaknesses

1. Assumption B.1 (Gaussian posteriors with shared variance) is strong and unverified for real diffusion model posteriors. No empirical validation is provided.
2. D_irr, the central theoretical quantity, is never computed or estimated for any model pair. Its actual magnitude and variation across architectures remain unknown.
3. The connection between the rate distortion bound (Theorem 4.2) and the geometric detection metrics (SAD, LGI) is conceptual rather than formally derived. These are presented as complementary perspectives rather than a unified deductive chain.

---

> ### Author Rebuttal · Authors · 2026-03-28
>
> Dear Reviewer CudM,
>
> Thank you for your thoughtful and detailed review. Your comments are very helpful in improving our work. We address each of your points in detail below, and we hope our responses resolve your concerns.
>
> > Weakness #1 & Question #2: Validation of Assumption B.1.
>
> We thank the reviewer for this point. We acknowledge that the Gaussian assumption and shared covariance structure are introduced to enable a tractable RD analysis, and may not fully capture the exact behavior of real diffusion posteriors.
>
> We emphasize that Assumption B.1 serves as a **conservative analytical baseline**. In practice, if the shared variance assumption is relaxed to account for the heterogeneous uncertainty structures, the KL divergence between the true and proxy posteriors is expected to increase. Thus, the derived distortion floor $D_{\mathrm{irr}}$ represents a conservative lower bound; the actual geometric discrepancy faced by an attacker is likely even larger.
>
> **Empirical Validation**: We provide empirical evidence to support these approximations:
>
> * **Gaussianity**: Using the TR scheme as an example, we perform a random projection normality test on the inversion residuals ($\hat{z}\_{T}^{(f)} - z\_T^{(w)}$) under the reprompting forgery attack. As shown in Table 1 below, across various target models, the average skewness and kurtosis are consistently **close to 0 and 3**, respectively.
>
>     Table 1. Normality test of inversion residuals (Proxy: SD2.1).
>
>     | Target | Skewness | Kurtosis |
>     | :--- | :---: | :---: |
>     | SD2.1 | 0.0092 | 3.1694 |
>     | SDXL | 0.0015 | 3.0272 |
>     | PixArt-Σ | -0.0083 | 3.0050 |
>     | FLUX | -0.0040 | 3.0612 |
>     | SD3 | -0.0106 | 3.0539 |
>
> * **Shared Variance**: To justify Assumption B.2, we compare the latent posterior variances in cross-architecture scenarios. The measured variance ratios are **1.2265** for SDXL/SD2.1 and **0.9508** for PixArt-Σ/SD2.1 (**both near-ideal 1.0**), with significant per-dimension correlations (0.4421 and 0.3496, respectively).
>
> Overall, these results confirm that our theoretical framework is a tractable first-order model and indicate that uncertainty structures are statistically consistent across different diffusion models. We will include this analysis in the revised manuscript.
>
> > Weakness #2 & Question #1: Quantifying Irreducible Distortion ($D_\mathrm{irr}$)
>
> We thank the reviewer for this comment. As discussed in Sec. B.2, we adopt the MSE as the distortion measure, which allows the RD function to admit a closed-form expression under the Gaussian latent assumption.
>
> The reconstruction error can be decomposed into **the intrinsic error introduced by the target model** and **the additional error induced by proxy-target mismatch**, where the latter error corresponds to $D_\mathrm{irr}$.
>
> Empirically, $D_\mathrm{irr}$ is estimated as the MSE gap between the proxy-based reconstruction and the intrinsic baseline (without mismatch). To validate this, we derive the distortion gap from the MSE values reported in Table 5 of the paper. The results for the TR scheme are summarized below:
>
> Table 2. Estimation of $D_\mathrm{irr}$ across different target-proxy pairs.
>
> | Target | Proxy: SD2.1 | ​Proxy: SD3 |
> | :--- | :---: | :---: |
> | **SD2.1** | **0.368** | 1.032 |
> | SDXL | 0.817 | 0.824 |
> | PixArt-Σ | 0.771 | 0.784 |
> | FLUX | 1.106 | 0.900 |
> | **SD3** | 0.923 | **0.432** |
>
> Notably, the estimated $D_\mathrm{irr}$ remains small when the proxy and target are identical, but increases significantly under cross-family settings. This confirms that $D_\mathrm{irr}$ is **measurable** and grows with model mismatch, consistent with our formulation. We will include this analysis in Sec. 5.4 of the revision to further bridge our theory and empirical results.
>
> > Comment #1: Connection between Removal Guarantees and Distortion Bounds.
>
> We are grateful to the reviewer for this constructive insight. While our work and Zhao et al. (2023) focus on different watermarking paradigms (**post-hoc vs. in-generation**), they can be viewed as the two **complementary ends** of a shared distortion continuum in the broader context of generative watermarking:
>
> (1) **Shared distortion continuum**. Zhao et al. show that an attacker must introduce sufficient stochastic perturbation to remove the watermark. In contrast, our work establishes a lower bound for forgery: even under faithful reconstruction, proxy-target mismatch imposes a deterministic floor $D_\mathrm{irr}$. These results define complementary limits of watermarking attacks along a common distortion spectrum.
>
> (2) **Unified information-theoretic view**. Both works implicitly characterize limits on the information that can be preserved or removed between the representation and the watermark signal, reflecting a fundamental trade-off between watermark removability and reconstruction fidelity.
>
> Per the reviewer’s suggestion, we will include a discussion of these connections in the revised manuscript.

---

> > ### Author Rebuttal · Reviewer_CudM · 2026-04-04
> >
> > My main concerns have been directly addresed. The Gaussian assumption is now empirically validated. D_irr has been estimated for specific model pairs, which concretizes the theoretical contribution and shows it varies meaningfully across architectures. The connections between removal guarantees and distortion bounds was also addressed thoughtfully. Therefore, I raise my score from 5 to 6.

---

> > > ### Author Response · Authors · 2026-04-07
> > >
> > > Dear Reviewer CudM,
> > >
> > > We sincerely thanks for your positive response and your feedback which helps improve our paper.

---

### Official Review · Reviewer_TUjn · 2026-03-13

**Soundness:** 2
**Presentation:** 1
**Significance:** 3
**Originality:** 3
**Overall Recommendation:** 3
**Confidence:** 4

**Summary:**

The paper studies black-box forgery attacks on semantic watermarks in diffusion models and argues that attack success is fundamentally limited when the attacker uses a proxy model that does not match the target model. Its main conceptual contribution is a rate–distortion view of forgery, where proxy–target mismatch creates an irreducible latent distortion floor that prevents perfect high-fidelity forgery. Building on this, the paper models forged latents as showing global drift and local deformation on the latent manifold, and proposes a scheme-agnostic pre-verification detector based on these geometric signals, with strong results across different black-box settings.

**Compliance With Llm Reviewing Policy:**

Affirmed.

**Final Justification:**

The paper addresses an important and timely problem, and the empirical study is broad and interesting. However, my main concern remains the **central theoretical claim**. The paper states a strong theorem about an irreducible distortion floor under posterior mismatch, but the derivation does not fully justify the key step from posterior mismatch to the claimed distortion lower bound. In my view, this part still reads more like an intuitive argument than a rigorous proof. A related issue is that the theory relies on a **simplified Gaussian/shared-covariance assumption** across proxy and target models. The paper itself presents this as a tractable modeling assumption, and I do not think the current empirical checks are strong enough to validate it at the level needed to support the theorem across heterogeneous diffusion backbones. So while I find the problem setting and empirical direction promising, I do not think the current version fully supports its main theoretical framing.

**Key Questions For Authors:**

See Cons

**Limitations:**

See Cons

**Strengths And Weaknesses:**

Pros:
1. The paper's strength is that it doesn't just say "our attack worked"—it thoughtfully explains why black-box forgery gets harder with proxy-target mismatch, using rate-distortion theory and linking it to geometric signals in latent space.
2. The empirical study is broad and credible: it tests six watermarking schemes, guidance/optimization forgeries, multiple targets, and 14 distortions.

Cons:
1. The central theorem argues that model mismatch creates an irreducible distortion floor and reduces the attacker’s effective information budget, but the proof feels more like an argument sketch than a tightly derived theorem. The jump from posterior mismatch to guaranteed distortion is especially under-supported.
2. The paper’s novelty comes more from combining existing ideas in a clever way than from introducing a fundamentally new watermarking primitive or a completely new security framework.
3. The paper is understandable at a high level, but several parts of the presentation are hard to follow.
4. The appendix relies on simplified Gaussian-style assumptions and shared uncertainty structure to make the analysis work. Those assumptions may be mathematically convenient, but they are not obviously faithful to the real behavior of inversion errors across different diffusion backbones.

---

> ### Author Rebuttal · Authors · 2026-03-28
>
> Dear Reviewer TUjn,
>
> Thank you for your thoughtful and detailed review. Your comments are very helpful in improving our work. We address each of your points in detail below, and we hope our responses resolve your concerns.
>
> > Weakness #1 Rigor of the Theoretical Derivation (Theorem 4.2)
>
> We thank the reviewer for this concern. We agree that the presentation can be made more explicit, particularly in connecting posterior mismatch to the distortion bound.
>
> We clarify that our result follows a standard RD interpretation under model mismatch. The discrepancy between the true posterior and the proxy posterior is quantified by the KL divergence $D_\mathrm{irr}$, which captures **an irreducible information loss**. Under the Gaussian latent assumption (Assumption B.1), this divergence induces an **effective rate penalty**, which directly leads to a lower bound on achievable distortion through the classical RD function.
>
> Intuitively, posterior mismatch not only introduces estimation noise but also reduces the information available to the adversary for reconstructing the latent. This can be interpreted as a rate penalty, which **shifts the RD curve** and induces a non-vanishing distortion floor even at high rates.
>
> In the revision, we will provide a clearer step-by-step derivation and additional intuition in Appendix B.2.
>
> > Weakness #2: Novelty and Conceptual Contribution
>
> We thank the reviewer for this point. While the metrics used are standard in representation analysis, our contribution lies in **a theoretical formulation of black-box forgery via RD and geometric analysis**.
>
> By modeling black-box forgery as an RD problem, we reveal that proxy–target mismatch inevitably introduces structured distortions in the latent space, manifested as (i) global directional drift on the hypersphere and (ii) local second-order deformation on the SPD manifold, explaining why forged samples remain distinguishable.
>
> The use of cosine similarity and covariance-based distances follows naturally from these geometric effects. More broadly, our framework is metric-agnostic, and alternative measures that capture **similar geometric discrepancies can be readily incorporated**. In addition, our method is computationally lightweight and satisfies the zero-storage requirement, making it practical for real-world deployment.
>
> Overall, the contribution is the theoretical characterization of forgery-induced geometric discrepancy and a scheme-agnostic pre-verification framework, rather than in the design of a new metric.
>
> > Weakness #3: Presentation clarity.
>
> We thank the reviewer for pointing out the presentation issue. In the revision, we will simplify Sec. 4, **add intuitive remarks** after theorems, and provide a notation summary in the Appendix. We will also improve clearer figures to illustrate the geometric interpretation. We would be happy to further clarify any specific parts if the reviewer finds them unclear.
>
> > Weakness #4: Concern about assumption justification.
>
> We thank the reviewer for this point. We acknowledge that the Gaussian assumption and shared covariance structure are introduced to enable a tractable RD analysis, and may not fully capture the exact behavior of real diffusion posteriors.
>
> We emphasize that Assumption B.1 serves as a **conservative analytical baseline**. In practice, if the shared variance assumption is relaxed to account for the heterogeneous uncertainty structures, the KL divergence between the true and proxy posteriors is expected to increase. Thus, the derived distortion floor $D_{\mathrm{irr}}$ represents a conservative lower bound; the actual geometric discrepancy faced by an attacker is likely even larger.
>
> **Empirical Validation**: We provide empirical evidence to support these approximations:
>
> * **Gaussianity**: Using the TR scheme as an example, we perform a random projection normality test on the inversion residuals ($\hat{z}\_{T}^{(f)} - z\_{T}^{(w)}$) under the reprompting forgery attack. As shown in Table 1 below, across various target models, the average skewness and kurtosis are consistently **close to 0 and 3**, respectively.
>
>     Table 1. Normality test of inversion residuals (Proxy: SD2.1).
>
>     | Target | Skewness | Kurtosis |
>     | :--- | :---: | :---: |
>     | SD2.1 | 0.0092 | 3.1694 |
>     | SDXL | 0.0015 | 3.0272 |
>     | PixArt-Σ | -0.0083 | 3.0050 |
>     | FLUX | -0.0040 | 3.0612 |
>     | SD3 | -0.0106 | 3.0539 |
>
> * **Shared Variance**: To justify Assumption B.2, we compare the latent posterior variances in cross-architecture scenarios. The measured variance ratios are **1.2265** for SDXL/SD2.1 and **0.9508** for PixArt-Σ/SD2.1 (**both near-ideal 1.0**), with significant per-dimension correlations (0.4421 and 0.3496, respectively).
>
> Overall, these results confirm that our theoretical framework is a tractable first-order model and indicate that uncertainty structures are statistically consistent across different diffusion models. We will include this analysis in the revised manuscript.

---

> > ### Author Rebuttal · Reviewer_TUjn · 2026-04-03
> >
> > Thank you for the detailed rebuttal.The response alleviates some of my concerns, but it does not sufficiently change my overall assessment. First, regarding the main theoretical claim, the rebuttal provides useful intuition for why proxy–target mismatch should induce reconstruction error, but it still does not fully justify the central jump from posterior mismatch to a guaranteed distortion floor. In particular, the argument remains closer to an RD-based interpretation than a tightly derived theorem, and the current response mainly promises a clearer derivation in a future revision rather than resolving the present concern. Second, regarding the appendix assumptions, the additional empirical checks are helpful, but they do not yet convincingly validate the Gaussian/shared-uncertainty assumptions at the level needed to support the stated theoretical claims across different diffusion backbones.The promised presentation improvements are also welcome,  but they do not address the core issue above. For these reasons, I decide to keep my original score unchanged.

---

> > > ### Author Response · Authors · 2026-04-03
> > >
> > > Dear Reviewer TUjn,
> > >
> > > Thank you for the insightful follow-up. We appreciate the opportunity to further solidify the theoretical connection and the validity of our underlying assumptions.
> > >
> > > > Re: Weakness #1 Formal Derivation of Theorem 4.2
> > >
> > > We thank the reviewer for this important point. To resolve the "jump" from posterior mismatch to the distortion floor, we provide a formal derivation via Pinsker's Inequality, which will replace the intuitive remarks in Sec 4.2.
> > >
> > > *Proof*: We formally derive the strictly positive lower bound of the distortion floor.
> > >
> > > **Step 1 (Bounding statistical distance via Pinsker’s Inequality).** Let $P = P_\Theta(X|Y_w)$ be the true target posterior and $Q=P_{\Theta_{\mathcal{A}}}(X|Y_w)$ be the proxy posterior, with a proxy-induced mismatch $D_{KL}(P||Q) = \epsilon\equiv D_{irr}$. By Pinsker's Inequality, the total variation distance between $P$ and $Q$ satisfies $\|P-Q\|\_{\mathrm{TV}}\leq \sqrt{2 D_{KL}(P||Q)}=\sqrt{2\epsilon}$, which quantifies the deviation between the two posteriors.
> > >
> > > **​Step 2 (Information degradation under posterior mismatch).** Any reconstruction $\hat{x}$ generated by the black-box adversary depends on samples from the mismatched proxy $Q$. The discrepancy between $P$ and $Q$ induces a degradation in the achievable mutual information under the true distribution. In particular, by standard continuity bounds of entropy and mutual information with respect to total variation distance, there exists a constant $\alpha > 0$ such that the effective rate satisfies $R_{\mathrm{eff}}\leq R_{\max}-\alpha \|P-Q\|\_{\mathrm{TV}}$. Combined with Step 1, we obtain $R_{\mathrm{eff}}\leq R_{\max}-\alpha \sqrt{2\epsilon}.$
> > >
> > > **Step 3 (Distortion lower bound via RD function).**  By Definition 3.1, under our Assumptions B.1 and B.2 (Gaussian source with variance $\sigma^2$ and MSE distortion), the inverse RD function is given by $D(R)=\sigma^2 2^{-2R}$.
> > >
> > > Substituting the penalized rate $R_{\mathrm{eff}}$ gives $\mathbb{E}\big[\|x-\hat{x}\|^2\big]\geq \sigma^2 \cdot 2^{-2\left(R_{\max}-\alpha \sqrt{2\epsilon}\right)}.$
> > >
> > > Crucially, the term $\alpha \sqrt{2\epsilon}$ provides a concrete realization of the information penalty $I_{\mathrm{pen}}$ defined in Theorem 4.2. It shows that any strict posterior mismatch ($\epsilon>0$) induces a strictly positive penalty on the effective rate. As a result, the expected distortion cannot vanish, establishing the existence of an irreducible distortion floor under the stated assumptions (Corollary 4.3).
> > >
> > > This completes the proof.
> > >
> > > We will incorporate this formal proof into the revised Sec. 4.2 and Appendix B.2, replacing the current discussion.
> > >
> > > > Re: Weakness #4 Theoretical Foundation of Assumptions
> > >
> > > We thank the reviewer for the rigorous critique. We provide the following justification for our Gaussian and shared-uncertainty assumptions across different backbones:
> > >
> > > **Theoretical Basis via Tweedie's Formula**: According to Tweedie’s Formula [1, 2], the posterior mean of clean data $x_0$ is determined by the score function: $$E[x_0\mid x_t]=\frac{1}{\sqrt{\alpha_t}}\left(x_t+(1-\alpha_t)\nabla \log p_t(x_t) \right).$$ Moreover, the posterior covariance is related to **the local curvature of the log-density**, i.e., $\nabla^2 \log p_t(x_t)$. These quantities are defined by **the data distribution and noise schedule**, rather than the specific model architecture. Since modern backbones (e.g., U-Net, DiT) are trained to approximate the same target score $\nabla\log p_t(x_t)$, their estimation error and uncertainty structures are statistically consistent, as supported by our earlier empirical results.
> > >
> > > **Remark (Deterministic Sampling and Practical Convergence)**: We note that modern diffusion/flow-matching frameworks (e.g., EDM, FLUX) increasingly adopt **straightened transport paths** and **deterministic sampling** (i.e., solving ODEs via Euler or Heun methods) for stable generation. Under such deterministic ODE trajectories, inversion and reconstruction are primarily governed by the learned vector field (drift) rather than stochastic noise injection, reducing sensitivity to the covariance structure. Therefore, in practice, this convergence mitigates residual variance discrepancies, supporting our shared-variance premise.
> > >
> > > Thus, Assumption B.1 is not merely a mathematically convenient simplification, but a realistic reflection of both the underlying score-matching objective and practical model standardization.
> > >
> > > References:
> > >
> > > [1] Efron, Bradley. "Tweedie’s formula and selection bias." Journal of the American Statistical Association 2011.
> > >
> > > [2] Luo, Calvin. "Understanding diffusion models: A unified perspective." arXiv preprint 2022.
> > >
> > > We appreciate your feedback throughout the review process, which has driven us to the rigor of our theoretical presentation. We hope the revised derivations and theoretical justifications address your concerns. If any points remain unclear, we would be happy to provide further clarification.

---

### Official Review · Reviewer_CCqk · 2026-03-13

**Soundness:** 3
**Presentation:** 3
**Significance:** 3
**Originality:** 3
**Overall Recommendation:** 4
**Confidence:** 1

**Summary:**

This paper focus on the problem of forgery attacks on semantic watermarking for diffusion models in a black-box setting. Attacker obtains a watermarked-image and performs inversion using a surrogate diffusion model to derive a latent representation. This latent is then used to generate  new images; while the content of the image change, the watermark will still persists, leading to source forgery. The paper claims that this attack is subject to theoretical limitations stemming from the discrepancy between the surrogate model and the target model. Utilizing rate-distortion theory, the authors explain that this discrepancy results in an unavoidable latent reconstruction error. Furthermore, the paper posits that this error is not merely random noise but manifests as geometric shifts in the latent space. These shifts are categorized into two types: global directional shifts and local structural deformations. To detect forged sample, the paper proposes two metrics: one is based on cos-similarity for global directional shifts, and another based on SPD covariance distance. Experiments across various diffusion models and semantic watermarking methods prove that this detection approach can effectively distinguish between real and forged samples in many scenario.

**Compliance With Llm Reviewing Policy:**

Affirmed.

**Final Justification:**

As I am not an expert in this field, I would kindly ask the AC to give less weight to my score and consider my evaluation accordingly.

**Key Questions For Authors:**

1. The paper’s method requires comparing the recovered latent against a reference. Is the method still viable in real world system where the original latent is not preserved?

2. The detection capability degrades significantly when the attacker uses the same diffusion model as the target. Has the paper considered stronger attacker scenarios, and are there strategy to maintain stable detection performance under such condition?

**Limitations:**

The proposed method relies on comparing the recovered latent with the reference latent generated by the target model. Since many real-world deployment scenarios do not store the latent information for every generated sample, the practical utility of this method is constrained.

**Strengths And Weaknesses:**

Contributions
1. Theoretical Analysis: The paper provides an information-theoretic analysis of forgery attacks on diffusion model semantic watermarking, identifying an unavoidable lower bound on distortion between the surrogate and target models.

2. Generalizable Detection: The paper proposes a forgery detection method based on latent geometric structures. Since it does not rely on specific watermarking algorithms, it is applicable to a wide range of semantic watermarking schemes.

Weaknesses
1. Strong Theoretical Assumptions: The paper assumes that diffusion latents follow an independent Gaussian distribution. However, latent distributions in real-world models are significantly more complex, creating a gap between the theoretical derivation and practical model behavior.

2. Limited Algorithmic Innovation: The proposed method is relatively straightforward; detecting forgery via latent cosine similarity and local covariance distance is common in representation learning and anomaly detection.

3. Weak Threat Model: The results indicate that the attack is almost entirely successful when the surrogate model matches the target model. The paper’s detection performance drops significantly in this scenario, yet the paper fails to address the reality that attackers may employ the same model architecture.

---

> ### Author Rebuttal · Authors · 2026-03-28
>
> Dear Reviewer CCqk,
>
> Thank you for your thoughtful and detailed review. Your comments are very helpful in improving our work. We address each of your points in detail below, and we hope our responses resolve your concerns.
>
> > Weakness #1: Strong Assumptions on diffusion models.
>
> We thank the reviewer for this point. In diffusion models, the initial latent is by definition sampled from a standard Gaussian distribution, and the training objective is defined under this assumption, a fundamental premise established in both theory and practice [1, 2].
>
> Our analysis follows this formulation to characterize latent geometry. Critically, the proposed detection remains robust for real-world latents that approximately follow this prior. Our experiments across a diverse set of models (SD2.1, SDXL, SD3, PixArt, and FLUX) demonstrate consistent geometric deviations, confirming that the method is robust to the slight distribution shifts encountered in practical latent spaces beyond the idealized setting.
>
> References:
>
> [1] Ho, Jonathan, et al. "Denoising diffusion probabilistic models." NeurIPS 2020.
>
> [2] Song, Yang, et al. "Score-Based Generative Modeling through Stochastic Differential Equations." ICLR 2021.
>
> > Weakness #2: Novelty and Conceptual Contribution.
>
> We thank the reviewer for this concern. While the metrics used are standard in representation analysis, our contribution lies in **a theoretical formulation of black-box forgery via RD and geometric analysis**.
>
> By modeling black-box forgery as an RD problem, we reveal that proxy–target mismatch inevitably introduces structured distortions in the latent space, manifested as (i) global directional drift on the hypersphere and (ii) local second-order deformation on the SPD manifold, explaining why forged samples remain distinguishable.
>
> The use of cosine similarity and covariance-based distances follows naturally from these geometric effects. More broadly, our framework is metric-agnostic, and alternative measures that capture **similar geometric discrepancies can be readily incorporated**. In addition, our method is computationally lightweight and satisfies the zero-storage requirement, making it practical for real-world deployment.
>
> Overall, the contribution is the theoretical characterization of forgery-induced geometric discrepancy and a scheme-agnostic pre-verification framework, rather than in the design of a new metric.
>
> > Weakness #3 & Question #2: Stronger attacker setting (same proxy–target models).
>
> We appreciate the reviewer’s insightful comment. We agree that the identical target-proxy setting is a critical security boundary, where geometric distortion becomes minimal, and detection is more challenging.
>
> To evaluate this scenario, we included **identical model** pairs in our experiments. As shown in Table 2 of the paper, even when the attacker uses the same architecture as the target (e.g., SD2.1 $\rightarrow$ SD 2.1), the method maintains **clear detection capability**. For example, on 1000 samples, the G-Cos metric achieves an AUC **above 0.955**, and L-SPD exceeds 0.928. For the SD3 family, the AUC remains **above 0.940 in most cases**. Furthermore, as demonstrated in Table 4 of the paper, this detection performance is **stable even under optimization-based attacks**.
>
> These results indicate that, under this strong attacker setting, forged samples still exhibit measurable geometric distortion in the recovered latent. Although the distortion is smaller than that of a model mismatch, it remains detectable, increasing the cost and uncertainty for the attacker and reducing the practical incentive to exploit semantic watermarks.
>
> In addition, our current L-SPD implementation uses fixed, task-agnostic hyperparameters (e.g., a 16×16 grid with top-5 mean aggregation). Under the TR scheme with reprompting attack in the same-model setting (SD2.1 $\rightarrow$ SD2.1), tuning these parameters (e.g., top-16 mean) elevates the AUC from **0.929** to **0.943**. This suggests that the proposed method is effective under fixed settings and can benefit from further parameter tuning. We will include a comprehensive ablation study in the revision.
>
> > Question #1 & Limitation #1: Concerns about storing reference latents.
>
> We thank the reviewer for raising this practical consideration. We clarify that our method introduces **zero storage overhead** (see Sec. 4.1& Sec. 6.3)
>
> In semantic watermarking (e.g., TR, GS), the initial watermarked latent ($z_T$) is deterministically generated from a secret key or seed, rather than being a random variable that must be saved. For **pre-verification**, the provider simply **regenerates** $z_T$ on-the-fly to compare with the recovered latent $\tilde{z}_T$. This **eliminates per-latent storage**, and we will further clarify this regeneration in the revision.

---

> > ### Author Rebuttal · Reviewer_CCqk · 2026-04-04
> >
> > Thank you for the response so far. It has partially alleviated my concerns.

---

> > > ### Author Response · Authors · 2026-04-04
> > >
> > > Thank you for your response. We would be grateful if you could indicate the unresolved concerns in more details, and we are glad to provide further clarification.

---

### Decision · Program_Chairs · 2026-04-30

**Decision:**

Accept (regular)

**Comment:**

This paper investigates black-box forgery attacks on semantic watermarks for diffusion models, making a significant contribution by framing the problem within a rate-distortion perspective. Reviewers generally agreed that the problem is timely and important, and several highlighted the paper’s strong empirical scope, including evaluations across multiple target and proxy models, watermarking schemes, attack types, and robustness settings. The paper’s main strengths are its principled attempt to explain when forgery attacks fundamentally fail, its geometric interpretation of the resulting distortion, and a scheme-agnostic pre-verification method that appears practically useful. Reviewers also noted the innovative motivation, solid presentation, and the combination of theoretical and empirical analysis.

The rebuttal was substantive and materially improved the paper. The authors added stronger justification for the Gaussian and shared-variance assumptions, clarified the stronger-attacker setting in which the adversary uses an identical proxy model, and added comparison against simpler alternative baselines, as well as quantitative estimates of the irreducible distortion floor. These additions resolved the concerns of two reviewers, both of whom raised their scores, and also strengthened the paper’s overall clarity and technical grounding. While one reviewer still preferred a tighter mathematical treatment and another remained only partially persuaded, the overall reviewer consensus after discussion is positive. Given the paper’s empirical support, theoretical perspective, and the improvements made during rebuttal, I recommend acceptance.